# How Do Vision Transformers Work?

**Namuk Park**[1,2]**, Songkuk Kim**[1]
[1]Yonsei University, [2]NAVER AI Lab
{namuk.park,songkuk}@yonsei.ac.kr

## Abstract

The success of multi-head self-attentions (MSAs) for computer vision is now indisputable. However, little is known about how MSAs work. We present fundamental explanations to help better understand the nature of MSAs. In particular, we demonstrate the following properties of MSAs and Vision Transformers (ViTs): ① MSAs improve not only accuracy but also generalization by flattening the loss landscapes. Such improvement is primarily attributable to their data specificity, not long-range dependency. On the other hand, ViTs suffer from non-convex losses. Large datasets and loss landscape smoothing methods alleviate this problem; ② MSAs and Convs exhibit opposite behaviors. For example, MSAs are low-pass filters, but Convs are high-pass filters. Therefore, MSAs and Convs are complementary; ③ Multi-stage neural networks behave like a series connection of small individual models. In addition, MSAs at the end of a stage play a key role in prediction. Based on these insights, we propose AlterNet, a model in which Conv blocks at the end of a stage are replaced with MSA blocks. AlterNet outperforms CNNs not only in large data regimes but also in small data regimes.

## 1 Introduction

There is limited understanding of multi-head self-attentions (MSAs), although they are now ubiquitous in computer vision. The most widely accepted explanation for the success of MSAs is their weak inductive bias and capture of long-range dependencies (See, e.g., (Dosovitskiy et al., 2021; Naseer et al., 2021; Tuli et al., 2021; Yu et al., 2021a; Mao et al., 2021; Chu et al., 2021)). Yet because of their over-flexibility, Vision Transformers (ViTs)—neural networks (NNs) consisting of MSAs—have been known to have a tendency to overfit training datasets, consequently leading to poor predictive performance in small data regimes, e.g., image classification on CIFAR. However, we show that the explanation is poorly supported.

### 1.1 Related Work

Self-attentions (Vaswani et al., 2017; Dosovitskiy et al., 2021) aggregate (spatial) tokens with normalized importances:

$$z_j = \sum_i \texttt{Softmax}\left(\frac{QK}{\sqrt{d}}\right)_i V_{i,j} \tag{1}$$

where $Q$, $K$, and $V$ are query, key, and value, respectively. $d$ is the dimension of query and key, and $z_j$ is the $j$-th output token. From the perspective of convolutional neural networks (CNNs), MSAs are a transformation of all feature map points with *large-sized* and *data-specific* kernels. Therefore, MSAs are at least as expressive as convolutional layers (Convs) (Cordonnier et al., 2020), although this does not guarantee that MSAs will behave like Convs.

Is the weak inductive bias of MSA, such as modeling long-range dependencies, beneficial for the predictive performance? To the contrary, appropriate constraints may actually help a model learn strong representations. For example, local MSAs (Yang et al., 2019; Liu et al., 2021; Chu et al., 2021), which calculate self-attention only within small windows, achieve better performance than global MSAs not only on small datasets but also on large datasets, e.g., ImageNet-21K.

In addition, prior works observed that MSAs have the following intriguing properties: ① MSAs improve the predictive performance of CNNs (Wang et al., 2018; Bello et al., 2019; Dai et al., 2021;

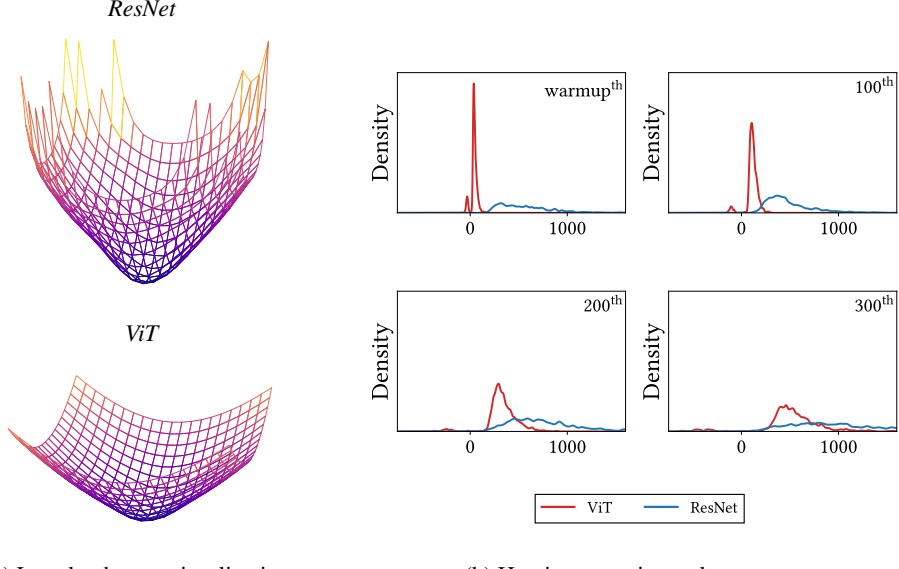

(a) Loss landscape visualizations          (b) Hessian max eigenvalue spectra

Figure 1: **Two different aspects consistently show that MSAs flatten loss landscapes**. *Left*: Loss landscape visualizations show that ViT has a flatter loss (NLL + $\ell_2$ regularization) than ResNet. *Right*: Hessian max eigenvalue spectra show that the magnitude of the Hessian eigenvalues of ViT is smaller than that of ResNet during training phases. We report the Hessian spectra at the end of the warmup phases, $100^{th}$, $200^{th}$, and $300^{th}$ epochs. See Fig. 4 for a more detailed analysis.

Guo et al., 2021; Srinivas et al., 2021), and ViTs predict well-calibrated uncertainty (Minderer et al., 2021). ② ViTs are robust against data corruptions, image occlusions (Naseer et al., 2021), and adversarial attacks (Shao et al., 2021; Bhojanapalli et al., 2021; Paul & Chen, 2022; Mao et al., 2021). They are particularly robust against high-frequency noises (Shao et al., 2021). ③ MSAs closer to the last layer significantly improve predictive performance (Graham et al., 2021; Dai et al., 2021).

These empirical observations raise immediate questions: ❶ What properties of MSAs do we need to better optimize NNs? Do the long-range dependencies of MSAs help NNs learn? ❷ Do MSAs act like Convs? If not, how are they different? ❸ How can we harmonize MSAs with Convs? Can we just leverage their advantages?

*We provide an explanation of how MSAs work by addressing them as a trainable spatial smoothing of feature maps*, because Eq. (1) also suggests that MSAs average feature map values with the positive importance-weights. Even non-trainable spatial smoothings, such as a small $2 \times 2$ box blur, help CNNs see better (Zhang, 2019; Park & Kim, 2021). These simple spatial smoothings not only improve accuracy but also robustness by spatially ensembling feature map points and flattening the loss landscapes (Park & Kim, 2021). Remarkably, spatial smoothings have the properties of MSAs ① – ③. See Appendix B for detailed explanations of MSAs as a spatial smoothing.

## 1.2 CONTRIBUTION

We address the three key questions:

❶ **What properties of MSAs do we need to improve optimization?** We present various evidences to support that MSA is generalized spatial smoothing. It means that MSAs improve performance because their formulation—Eq. (1)—is an appropriate inductive bias. Their weak inductive bias disrupts NN training. In particular, *a key feature of MSAs is their data specificity*, not long-range dependency. As an extreme example, local MSAs with a $3 \times 3$ receptive field outperforms global MSA because they reduce unnecessary degrees of freedom.

How do MSAs improve performance? MSAs have their advantages and disadvantages. On the one hand, they flatten loss landscapes as shown in Fig. 1. The flatter the loss landscape, the better the

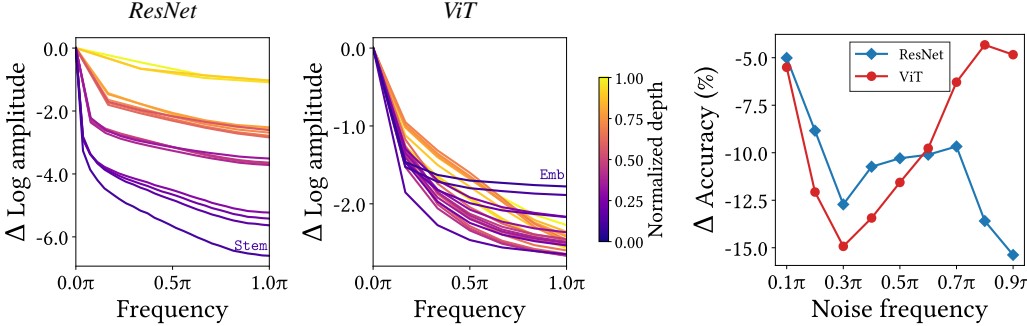

(a) Relative log amplitudes of Fourier transformed feature maps.

(b) Robustness for noise frequency

Figure 2: **The Fourier analysis shows that MSAs do not act like Convs**. *Left*: Relative log amplitudes of Fourier transformed feature map show that ViT tends to reduce high-frequency signals, while ResNet amplifies them. $\Delta$ Log amplitude is the difference between the log amplitude at normalized frequency $0.0\pi$ (center) and at $1.0\pi$ (boundary). See Fig. 8 for more detailed analysis. *Right*: We measure the decrease in accuracy against frequency-based random noise. ResNet is vulnerable to high-frequency noise, while ViT is robust against them. We use frequency window size of $0.1\pi$.

performance and generalization (Li et al., 2018; Keskar et al., 2017; Santurkar et al., 2018; Foret et al., 2021; Chen et al., 2022). Thus, they improve not only accuracy but also robustness in large data regimes. On the other hand, MSAs allow negative Hessian eigenvalues in small data regimes. This means that the loss landscapes of MSAs are non-convex, and this non-convexity disturbs NN optimization (Dauphin et al., 2014). Large amounts of training data suppress negative eigenvalues and convexify losses.

**②  Do MSAs act like Convs?**    We show that MSAs and Convs exhibit opposite behaviors. MSAs aggregate feature maps, but Convs diversify them. Moreover, as shown in Fig. 2a, the Fourier analysis of feature maps shows that MSAs reduce high-frequency signals, while Convs, conversely, amplifies high-frequency components. In other words, *MSAs are low-pass filters, but Convs are high-pass filters*. In addition, Fig. 2b indicates that Convs are vulnerable to high-frequency noise but that MSAs are not. Therefore, MSAs and Convs are complementary.

**③  How can we harmonize MSAs with Convs?**    We reveal that multi-stage NNs behave like a series connection of small individual models. Thus, applying spatial smoothing at the end of a stage improves accuracy by ensembling transformed feature map outputs from each stage (Park & Kim, 2021) as shown in Fig. 3a. Based on this finding, *we propose an alternating pattern of Convs and MSAs*. NN stages using this design pattern consists of a number of CNN blocks and one (or a few) MSA block as shown in Fig. 3c. The design pattern naturally derives the structure of canonical Transformer, which has one MSA block per MLP block as shown in Fig. 3b. It also provides an explanation of how adding Convs to Transformer's MLP block improves accuracy and robustness (Yuan et al., 2021; Guo et al., 2021; Mao et al., 2021).

Surprisingly, models using this alternating pattern of Convs and MSAs outperform CNNs not only on large datasets but also on small datasets, such as CIFAR. This contrasts with canonical ViTs, models that perform poorly on small amount of data. It implies that MSAs are generalized spatial smoothings that complement Convs, not simply generalized Convs.

## 2    What Properties of MSAs Do We Need To Improve Optimization?

To understand the underlying nature of MSAs, we investigate the properties of the ViT family: e.g., vanilla ViT (Dosovitskiy et al., 2021); PiT (Heo et al., 2021), which is "ViT + multi-stage"; and Swin (Liu et al., 2021), which is "ViT + multi-stage + local MSA". This section shows that these additional inductive biases enable ViTs to learn strong representations. We also use ResNet (He et al., 2016a) for comparison. NNs are trained from scratch with DeiT-style data augmentation (Touvron et al., 2021) for 300 epochs. The NN training begins with a gradual warmup (Goyal et al., 2017) for 5 epochs. Appendix A provides more detailed configurations and background information for experiments.

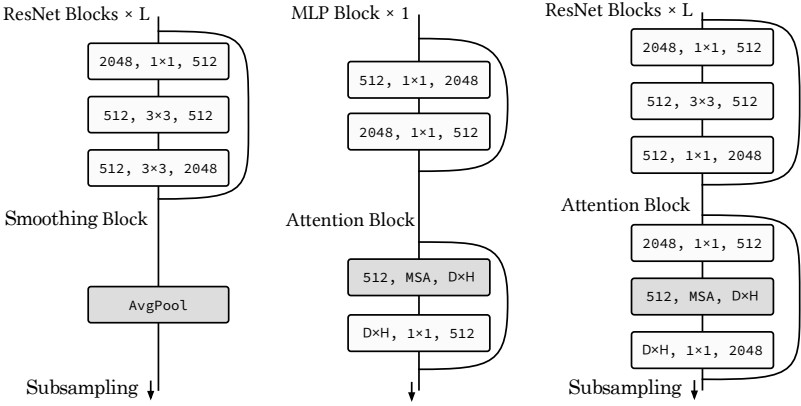

(a) Spatial smoothing     (b) Canonical Transformer   (c) Alternating pattern (*ours*)

Figure 3: **Comparison of three different repeating patterns**. *Left*: Spatial smoothings are located at the end of CNN stages. *Middle*: The stages of ViTs consist of repetitions of canonical Transformers. "D" is the hidden dimension and "H" is the number of heads. *Right*: The stages using alternating pattern consists of a number of CNN blocks and an MSA block. For more details, see Fig. 11.

**The stronger the inductive biases, the stronger the representations (*not regularizations*).** Do models with weak inductive biases overfit training datasets? To address this question, we provide two criteria on CIFAR-100: the error of the test dataset and the cross-entropy, or the negative log-likelihood, of the training dataset ($NLL_{train}$, the lower the better). See Fig. 5a for the results.

Contrary to our expectations, experimental results show that the stronger the inductive bias, the lower *both* the test error and the training NLL. This indicates that *ViT does not overfit training datasets*. In addition, appropriate inductive biases, such as locality constraints for MSAs, helps NNs learn strong representations. We also observe these phenomena on CIFAR-10 and ImageNet as shown in Fig. C.1. Figure C.2 also supports that weak inductive biases disrupt NN training. In this experiment, extremely small patch sizes for the embedding hurt the predictive performance of ViT.

**ViT does not overfit small training datasets.** We observe that ViT does not overfit even on smaller datasets. Figure 5b shows the test error and the training NLL of ViT on subsampled datasets. In this experiment, as the size of the dataset decreases, the error increases as expected, but surprisingly, $NLL_{train}$ also increases. Thanks to the strong data augmentation, ViT does not overfit even on a dataset size of 2%. This suggests that ViT's poor performance in small data regimes is *not* due to overfitting.

**ViT's non-convex losses lead to poor performance.** How do weak inductive biases of MSAs disturb the optimization? A loss landscape perspective provides an explanation: *the loss function of ViT is non-convex, while that of ResNet is strongly (near-)convex*. This poor loss disrupts NN training (Dauphin et al., 2014), especially in the early phase of training (Jastrzebski et al., 2020; 2021). Figure 1b and Fig. 4 provide top-5 largest Hessian eigenvalue densities (Park & Kim, 2021) with a batch size of 16. The figures show that ViT has a number of negative Hessian eigenvalues, while ResNet only has a few.

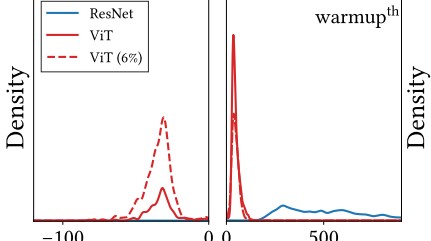

Figure 4: **Hessian max eigenvalue spectra show that MSAs have their advantages and disadvantages.** The dotted line is the spectrum of ViT using 6% dataset for training. *Left*: ViT has a number of negative Hessian eigenvalues, while ResNet only has a few. *Right*: The magnitude of ViT's positive Hessian eigenvalues is small. See also Fig. 1b for more results.

Figure 4 also shows that large datasets suppress negative Hessian eigenvalues in the early phase of training. Therefore, large datasets tend to help ViT learn strong representations by convexifying the loss. ResNet enjoys little benefit from large datasets because its loss is convex even on small datasets.

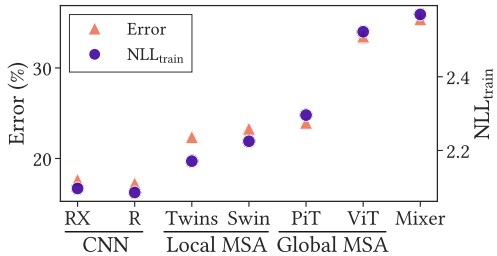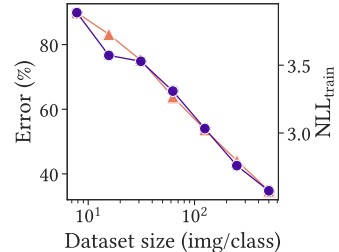

(a) Error and $\text{NLL}_{\text{train}}$ for each model.    (b) Performance of ViT for dataset size.

Figure 5: **ViT does not overfit training datasets.** "R" is ResNet and "RX" is ResNeXt. *Left:* Weak inductive bias disturbs NN optimization. The lower the $\text{NLL}_{\text{train}}$, the lower the error. *Right:* The lack of dataset also disturbs NN optimization.

**Loss landscape smoothing methods aids in ViT training.** Loss landscape smoothing methods can also help ViT learn strong representations. In classification tasks, global average pooling (GAP) smoothens the loss landscape by strongly ensembling feature map points (Park & Kim, 2021). We demonstrate how the loss smoothing method can help ViT improve performance by analyzing ViT with GAP classifier instead of CLS token on CIFAR-100.

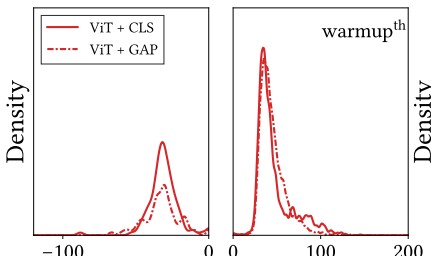

Figure 6 shows the Hessian max eigenvalue spectrum of the ViT with GAP. As expected, the result shows that GAP classifier suppresses negative Hessian max eigenvalues, suggesting that GAP convexify the loss. Since negative eigenvalues disturb NN optimization, GAP classifier improve the accuracy by $+2.7$ percent point.

Figure 6: **GAP classifier suppresses negative Hessian max eigenvalues** in an early phase of training. We present Hessian max eigenvalue spectrum of ViT with GAP classifier instead of CLS token.

Likewise, Sharpness-Aware Minimization (SAM) (Foret et al., 2021), an optimizer that relies on the local smoothness of the loss function, also helps NNs seek out smooth minima. Chen et al. (2022) showed that SAM improves the predictive performance of ViT.

**MSAs flatten the loss landscape.** Another property of MSAs is that they reduces the magnitude of Hessian eigenvalues. Figure 1b and Fig. 4 shows that the eigenvalues of ViT are significantly smaller than that of CNNs. While large eigenvalues impede NN training (Ghorbani et al., 2019), *MSAs can help NNs learn better representations by suppressing large Hessian eigenvalues.* Figure 1a also support this claim. In Fig. 1a, we visualize the loss landscapes by using filter normalization (Li et al., 2018), and the loss landscape of ViT is flatter than that of ResNet. In large data regimes, the negative Hessian eigenvalues—the disadvantage of MSAs—disappears, and only their advantages remain. As a result, ViTs outperform CNNs on large datasets, such as ImageNet and JFT (Sun et al., 2017). PiT and Swin also flatten the loss landscapes. For more details, see Fig. C.4.

**A key feature of MSAs is data specificity** *(not long-range dependency).* The two distinguishing features of MSAs are long-range dependency and data specificity, also known as data dependency, as discussed in Section 1.1. Contrary to popular belief, the long-range dependency hinders NN optimization. To demonstrate this, we analyze *convolutional ViT*, which consists of two-dimensional convolutional MSAs (Yang et al., 2019) instead of global MSAs. Convolutional MSAs calculates self-attention only between feature map points in convolutional receptive fields after unfolding the feature maps in the same way as convolutions.

Figure 7a shows the error and $\text{NLL}_{\text{train}}$ of convolutional ViTs with kernel sizes of $3 \times 3$, $5 \times 5$, and $8 \times 8$ (global MSA) on CIFAR-100. In this experiment, $5 \times 5$ kernel outperforms $8 \times 8$ kernel on both the training and the test datasets. $\text{NLL}_{\text{train}}$ of $3 \times 3$ kernel is worse than that of $5 \times 5$ kernel, but better than that of global MSA. Although the test accuracies of $3 \times 3$ and $5 \times 5$ kernels are comparable, the robustness of $5 \times 5$ kernel is significantly better than that of $3 \times 3$ kernel on CIFAR-100-C (Hendrycks & Dietterich, 2019).

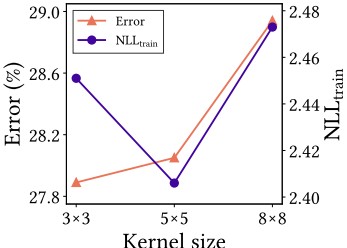 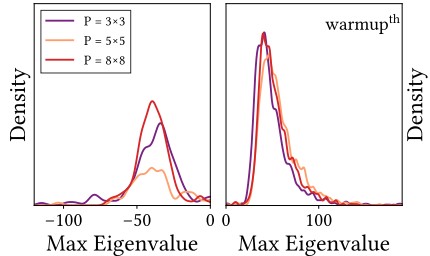

(a) Error and NLL$_{\text{train}}$ of ViT with local MSA for kernel size

(b) Hessian negative and positive max eigenvalue spectra in early phase of training

Figure 7: **Locality constraint improves the performance of ViT**. We analyze the ViT with convolutional MSAs. Convolutional MSA with $8 \times 8$ kernel is global MSA. *Left:* Local MSAs learn stronger representations than global MSA. *Right:* Locality inductive bias suppresses the negative Hessian eigenvalues, i.e., local MSAs have convex losses.

Figure 7b shows that the strong locality inductive bias not only reduce computational complexity as originally proposed (Liu et al., 2021), but also aid in optimization by convexifying the loss landscape. $5 \times 5$ kernel has fewer negative eigenvalues than global MSA because it restricts unnecessary degrees of freedom. $5 \times 5$ kernel also has fewer negative eigenvalues than $3 \times 3$ kernel because it ensembles a larger number of feature map points (See also Fig. 6). The amount of negative eigenvalues is minimized when these two effects are balanced.

It is clear that data specificity improves NNs. MLP-Mixer (Tolstikhin et al., 2021; Yu et al., 2021a), a model with an MLP kernel that does not depend on input data, underperforms compared to ViTs. Data specificity without self-attention (Bello, 2021) improves performance.

## 3 DO MSAS ACT LIKE CONVS?

Convs are data-agnostic and channel-specific. In contrast, MSAs are data-specific and channel-agnostic. This section shows that these differences lead to large behavioral differences. It suggests that MSAs and Convs are complementary.

**MSAs are low-pass filters, but Convs are high-pass filters.** As explained in Section 1.1, MSAs spatially smoothen feature maps with self-attention importances. Therefore, we expect that MSAs will tend to reduce high-frequency signals. See Appendix B for a more detailed discussion.

Figure 8 shows the relative log amplitude ($\Delta$ log amplitude) of ViT's Fourier transformed feature map at high-frequency ($1.0\pi$) on ImageNet. In this figure, MSAs almost always decrease the high-frequency amplitude, and MLPs—corresponding to Convs—increase it. The only exception is in the early stages of the model. In these stages, MSAs behave like Convs, i.e., they increase the amplitude. This could serve as an evidence for a hybrid model that uses Convs in early stages and MSAs in late stages (Guo et al., 2021; Graham et al., 2021; Dai et al., 2021; Xiao et al., 2021; Srinivas et al., 2021).

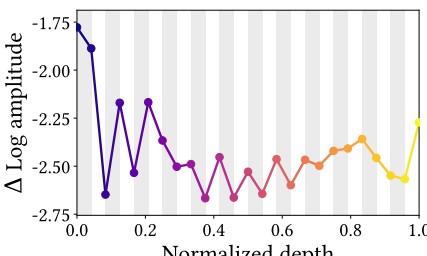

Figure 8: **MSAs (gray area) generally reduce the high-frequency component of feature map, and MLPs (white area) amplify it**. This figure provides $\Delta$ log amplitude of ViT at high-frequency ($1.0\pi$). See also Fig. 2a and Fig. D.2 for more results.

Based on this, we can infer that *low-frequency signals and high-frequency signals are informative to MSAs and Convs, respectively*. In support of this argument, we report the robustness of ViT and ResNet against frequency-based random noise. Following Shao et al. (2021) and Park & Kim (2021), we measure the decrease in accuracy with respect to data with frequency-based random noise $x_{\text{noise}} = x_0 + \mathcal{F}^{-1}\left(\mathcal{F}(\delta) \odot \mathbf{M}_f\right)$, where $x_0$ is clean data, $\mathcal{F}(\cdot)$ and $\mathcal{F}^{-1}(\cdot)$ are Fourier transform and inverse Fourier transform, $\delta$ is Gaussian random noise, and $\mathbf{M}_f$ is frequency mask.

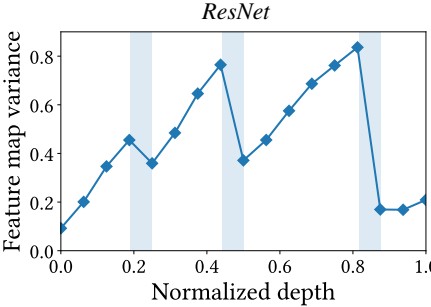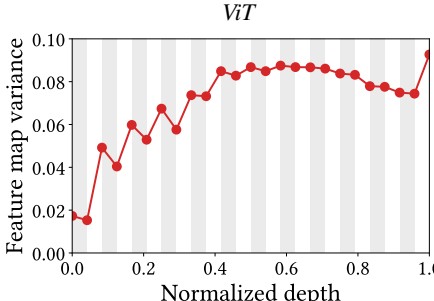

Figure 9: **MSAs (gray area) reduce the variance of feature map points, but Convs (white area) increase the variance.** The blue area is subsampling layer. This result implies that MSAs ensemble feature maps, but Convs do not.

As expected, the result in Fig. 2b reveals that ViT and ResNet are vulnerable to low-frequency noise and high-frequency noise, respectively. Low-frequency signals and the high-frequency signals each correspond to the shape and the texture of images. The results thus suggests that MSAs are shape-biased (Naseer et al., 2021), whereas Convs are texture-biased (Geirhos et al., 2019).

**MSAs aggregate feature maps, but Convs do not.** Since MSAs average feature maps, they will reduce variance of feature map points. This suggests that MSAs ensemble feature maps (Park & Kim, 2021). To demonstrate this claim, we measure the variance of feature maps from NN layers.

Figure 9 shows the experimental results of ResNet and ViT. This figure indicates that MSAs in ViT tend to reduce the variance; conversely, Convs in ResNet and MLPs in ViT increase it. In conclusion, *MSAs ensemble feature map predictions, but Convs do not*. As Park & Kim (2021) figured out, reducing the feature map uncertainty helps optimization by ensembling and stabilizing the transformed feature maps. See Fig. D.1 for more results on PiT and Swin.

We observe two additional patterns for feature map variance. First, the variance accumulates in every NN layer and tends to increase as the depth increases. Second, the feature map variance in ResNet peaks at the ends of each stage. Therefore, we can improve the predictive performance of ResNet by inserting MSAs at the end of each stage. Furthermore, we also can improve the performance by using MSAs with a large number of heads in late stages.

## 4 How Can We Harmonize MSAs With Convs?

Since MSAs and Convs are complementary, this section seeks to design a model that leverages only the advantages of the two modules. To this end, we propose the design rules described in Fig. 3c, and demonstrate that the models using these rules outperforms CNNs, not only in the large data regimes but also in the small data regimes, such as CIFAR.

### 4.1 Designing Architecture

We first investigate the properties of multi-stage NN architectures. Based on this investigation, we come to propose an alternating pattern, i.e., a principle for stacking MSAs based on CNNs.

**Multi-stage NNs behave like individual models.** In Fig. 9, we observe that the pattern of feature map variance repeats itself at every stages. This behavior is also observed in feature map similarities and lesion studies.

Figure 10a shows the representational similarities of ResNet and Swin on CIFAR-100. In this experiment, we use mini-batch CKA (Nguyen et al., 2021) to measure the similarities. As Nguyen et al. (2021) figured out, the feature map similarities of CNNs have a block structure. Likewise, we observe that the feature map similarities of multi-stage ViTs, such as PiT and Swin, also have a block structure. Since vanilla ViT does not have this structure (Bhojanapalli et al., 2021; Raghu et al., 2021), the structure is an intrinsic characteristic of multi-stage architectures. See Fig. D.3 for more detailed results of ViT and PiT.

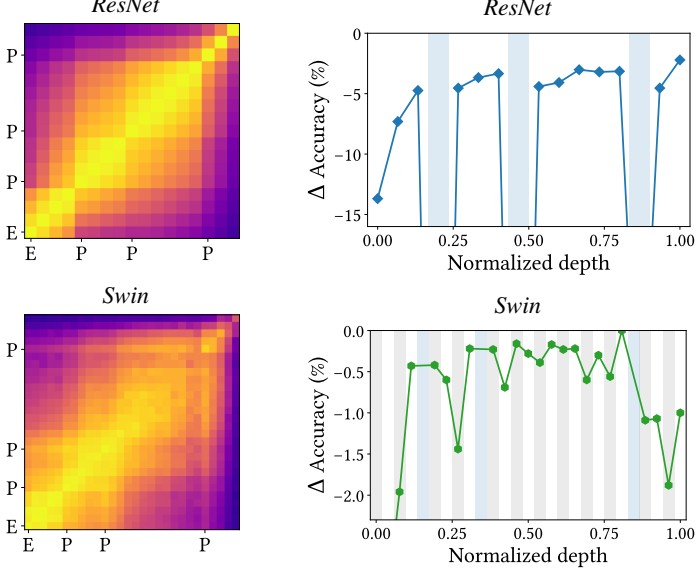

(a) Feature map similarity     (b) Accuracy of one-unit-removed model.

Figure 10: **Multi-stage CNNs and ViTs behave like a series connection of small individual models.** *Left:* The feature map similarities show the block structure of ResNet and Swin. "E" stands for stem/embedding and "P" for pooling (subsampling) layer. *Right:* We measure decrease in accuracy after removing one unit from the trained model. Accuracy changes periodically, and this period is one stage. White, gray, and blue areas are Conv/MLP, MSA, and subsampling layers, respectively.

Figure 10b shows the results of lesion study (Bhojanapalli et al., 2021), where one NN unit is removed from already trained ResNet and Swin during the testing phase. In this experiment, we remove one $3 \times 3$ Conv layer from the bottleneck block of ResNet, and one MSA or MLP block from Swin. In ResNet, removing an early stage layers hurts accuracy more than removing a late stage layers. More importantly, removing a layer at the beginning of a stage impairs accuracy more than removing a layer at the end of a stage. The case of Swin is even more interesting. At the beginning of a stage, removing an MLP hurts accuracy. At the end of a stage, removing an MSA seriously impairs the accuracy. These results are consistent with Fig. 8. See Fig. D.4 for the results on ViT and PiT.

Based on these findings, we expect MSAs closer to the end of *a stage* to significantly improve the predictive performance. This is contrary to the popular belief that MSAs closer to the end of a model improve the performance (Srinivas et al., 2021; d'Ascoli et al., 2021; Graham et al., 2021; Dai et al., 2021).

**Build-up rule.** Considering all the insights, we propose the following design rules:

- Alternately replace Conv blocks with MSA blocks from the end of a baseline CNN model.

- If the added MSA block does not improve predictive performance, replace a Conv block located at the end of an earlier stage with an MSA block .

- Use more heads and higher hidden dimensions for MSA blocks in late stages.

We call the model that follows these rules *AlterNet*. AlterNet unifies ViTs and CNNs by adjusting the ratio of MSAs and Convs as shown in Fig. 3. Figure 11 shows AlterNet based on pre-activation ResNet-50 (He et al., 2016b) for CIFAR-100 as an example. Figure D.5 shows AlterNet for ImageNet.

Figure 12a reports the accuracy of *Alter-ResNet-50*, which replaces the Conv blocks in ResNet-50 with local MSAs (Liu et al., 2021) according to the aforementioned rules, on CIFAR-100. As expected, MSAs in the last stage (c4) significantly improve the accuracy. Surprisingly, an MSA in 2nd stage (c2) improves the accuracy, while two or more MSAs in the 3rd stage (c3) reduce it. In conclusion, MSAs at the end of a stage play an important role in prediction.

Figure 12c demonstrates that MSAs suppress large eigenvalues while allowing only a few negative eigenvalues. As explained in Fig. 4, large datasets compensate for the shortcomings of MSAs. Therefore, more data allows more MSAs for a models.

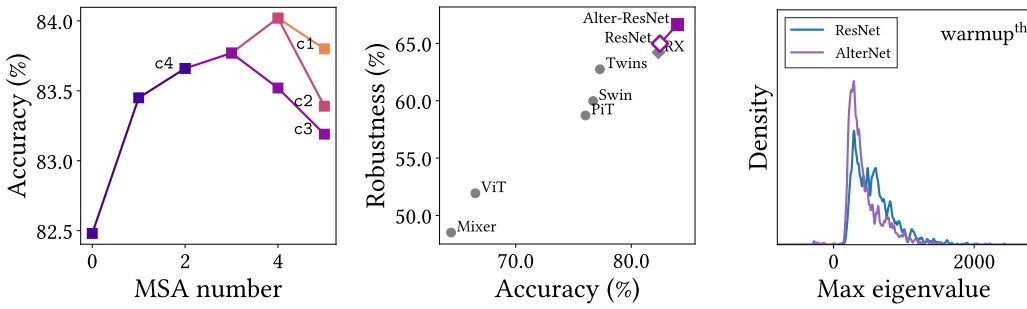

Figure 11: **Detailed architecture of Alter-ResNet-50 for CIFAR-100.** White, gray, and blue blocks mean Conv, MSA, and subsampling blocks. All stages (except stage 1) end with MSA blocks. This model is based on pre-activation ResNet-50. Following Swin, MSAs in stages 1 to 4 have 3, 6, 12, and 24 heads, respectively.

(a) Accuracy of AlterNet for MSA number

(b) Accuracy and robustness in a small data regime (CIFAR-100)

(c) Hessian max eigenvalue spectra in an early phase of training

Figure 12: **AlterNet outperforms CNNs and ViTs.** *Left:* MSAs in the late of the stages improve accuracy. We replace Convs of ResNet with MSAs one by one according to the build-up rules. c1 to c4 stands for the stages. Several MSAs in c3 harm the accuracy, but the MSA at the end of c2 improves it. *Center:* AlterNet outperforms CNNs even in a small data regime. Robustness is mean accuracy on CIFAR-100-C. "RX" is ResNeXt. *Right:* MSAs in AlterNet suppress the large eigenvalues; i.e., AlterNet has a flatter loss landscape than ResNet in the early phase of training.

## 4.2 PERFORMANCE

Figure 12b shows the accuracy and corruption robustness of Alter-ResNet-50 and other baselines on CIFAR-100 and CIFAR-100-C. Since CIFAR is a small dataset, CNNs outperforms canonical ViTs. Surprisingly, Alter-ResNet—a model with MSAs following the appropriate build-up rule—outperforms CNNs even in the small data regimes. This suggests that MSAs complement Convs. In the same manner, this simple modification shows competitive performance on larger datasets, such as ImageNet. See Fig. E.1 for more details.

## 5 DISCUSSION

Our present work demonstrates that MSAs are not merely generalized Convs, but rather generalized spatial smoothings that complement Convs. MSAs help NNs learn strong representations by ensembling feature map points and flattening the loss landscape.

Since the main objective of this work is to investigate the nature of MSA for computer vision, we preserve the architectures of Conv and MSA blocks in AlterNet. Thus, AlterNet has a strong potential for future improvements. In addition, AlterNet can conveniently replace the backbone for other vision tasks such as dense prediction (Carion et al., 2020). As Park & Kim (2021) pointed out, global average pooling (GAP) for simple classification tasks has a strong tendency to ensemble feature maps, but NNs for dense prediction do not use GAP. Therefore, we believe that MSA to be able to significantly improve the results in dense prediction tasks by ensembling feature maps. Lastly, strong data augmentation for MSA training harms uncertainty calibration as shown in Fig. F.1a. We leave a detailed investigation for future work.

## ACKNOWLEDGEMENT

We thank the reviewers, Taeoh Kim, and Pilhyeon Lee for valuable feedback. This work was supported by the Samsung Science and Technology Foundation under Project Number SSTF-BA1501-52.

## REPRODUCIBILITY STATEMENT

To ensure reproducibility, we provide comprehensive resources, such as code and experimental details. The code is available at https://github.com/xxxnell/how-do-vits-work. Appendix A.1 provides the specifications of all models used in this work. Detailed experimental setup including hyperparameters and the structure of AlterNet are also available in Appendix A.1 and Appendix E. De-facto image datasets are used for all experiments as described in Appendix A.1.

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

## A    Experimental Details

This section provides experimental details, e.g., setups and background information.

### A.1    Setups

We obtain the main experimental results from two sets of machines for CIFAR (Krizhevsky et al., 2009). The first set consists of an Intel Xeon W-2123 Processor, 32GB memory, and a single GeForce RTX 2080 Ti, and the other set of four Intel Intel Broadwell CPUs, 15GB memory, and a single NVIDIA T4. For ImageNet (Russakovsky et al., 2015), we use AMD Ryzen Threadripper 3960X 24-Core Processor, 256GB memory, and four GeForce RTX 2080 Ti. NN models are implemented in PyTorch (Paszke et al., 2019).

We train NNs using categorical cross-entropy (NLL) loss and AdamW optimizer (Loshchilov & Hutter, 2019) with initial learning rate of $1.25 \times 10^{-4}$ and weight decay of $5 \times 10^{-2}$. We also use cosine annealing scheduler (Loshchilov & Hutter, 2017). NNs are trained for 300 epochs with a batch size of 96 on CIFAR, and a batch size of 128 on ImageNet. The learning rate is gradually increased (Goyal et al., 2017) for 5 epochs. Following Touvron et al. (2021), strong data augmentations—such as RandAugment (Cubuk et al., 2020), Random Erasing (Zhong et al., 2020), label smoothing (Szegedy et al., 2016), mixup (Zhang et al., 2018), and CutMix (Yun et al., 2019)—are used for training. Stochastic depth (Huang et al., 2016) is also used to regularize NNs. This DeiT-style configuration, which significantly improves the performance (Steiner et al., 2021; Bello et al., 2021), is the de facto standard in ViT training (See, e.g., (Heo et al., 2021; Liu et al., 2021)). Therefore, we believe the insights presented in this paper can be used widely. See source code (`https://github.com/xxxnell/how-do-vits-work`) for detailed configurations.

We mainly report the performances of ResNet-50, ViT-Ti, PiT-Ti, and Swin-Ti. Their training throughputs on CIFAR-100 are 320, 434, 364, and 469 image/sec, respectively, which are comparable to each other. Figures 5a and C.1a report the predictive performance of ResNeXt-50 (Xie et al., 2017), Twins-S (Chu et al., 2021), and MLP-Mixer-Ti (Tolstikhin et al., 2021). Figure E.1 additionally reports the performance of ConViT-Ti (d'Ascoli et al., 2021), LeViT-128S (Graham et al., 2021), and CoaT-Lite-Ti (Xu et al., 2021). We use a patch size of $2 \times 2$ for ViT and PiT on CIFAR; for Swin, a patch size of $1 \times 1$ and a window size of $4 \times 4$. We use a patch size of $4 \times 4$ for ViT only in Fig. 7. We halve the depth of the ViT in Fig. C.5 and Fig. C.6 due to the memory limitation.

All models for CIFAR, and ResNet, ViT, and AlterNet for ImageNet are trained from scratch. We use pertained PiT and Swin from Wightman (2019) for ImageNet. The implementations of Vision Transformers are based on Wightman (2019) and Wang (2021).

For Hessian max eigenvalue spectrum (Park & Kim, 2021), 10% of the training dataset is used. We also use power iteration with a batch size of 16 to produce the top-5 largest eigenvalues. To this end, we use the implementation of Yao et al. (2020). We modify the algorithm to calculate the eigenvalues with respect to $\ell_2$ regularized NLL on augmented training datasets. In the strict sense, the weight decay is not $\ell_2$ regularization, but we neglect the difference.

For the Fourier analysis and the feature map variance experiment, the entire test dataset is used. We report the amplitudes and the variances averaged over the channels.

### A.2    Background Information

Below are the preliminaries and terms of our experiments.

**Test error and training NLL.**    We report test errors on clean test datasets and training NLLs on augmented training datasets in experiments, e.g., Fig. 5 and Fig. C.1. NLL is an appropriate metric for evaluating convergence on a training dataset because an NN optimizes NLL. In addition, it is the most widely used as a proper scoring rule indicating both accuracy and uncertainty. To represent predictive performance on a test dataset, we use a well-known metric: error. Although NLL can also serve the same purpose, results are consistent even when NLL is employed.

If an additional inductive bias or a learning technique improves the performance of an NN, this is either a method to help the NNs learn *"strong representations"*, or a method to *"regularize"* it.

An improved—i.e., lower—training NLL suggests that this bias or technique helps the NN learn strong representations. Conversely, a compromised training NLL indicates that the bias or technique regularizes the NN. Likewise, we say that *"an NN overfits a training dataset"* when a test error is compromised as the training NLL is improved.

**Hessian max eigenvalue spectrum.** Park & Kim (2021) proposed *"Hessian max eigenvalue spectra"*, a feasible method for visualizing Hessian eigenvalues of large-sized NNs for real-world problems. It calculates and gathers top-$k$ Hessian eigenvalues by using power iteration mini-batch wisely. Ghorbani et al. (2019) visualized the Hessian eigenvalue spectrum by using the Lanczos quadrature algorithm for full batch. However, this is not feasible for practical NNs because the algorithm requires a lot of memory and computing resources.

A good loss landscape is a flat and convex loss landscape. Hessian eigenvalues indicate the flatness and convexity of losses. The magnitude of Hessian eigenvalues shows sharpness, and the presence of negative Hessian eigenvalues shows non-convexity. Based on these insights, we introduce *a negative max eigenvalue proportion* (NEP, the lower the better) and *an average of positive max eigenvalues* (APE, the lower the better) to quantitatively measure the non-convexity and the sharpness, respectively. For a Hessian max eigenvalue spectrum $p(\lambda)$, NEP is the proportion of negative eigenvalues $\int_{-\infty}^{0} p(\lambda)\, d\lambda$, and APE is the expected value of positive eigenvalues $\int_{0}^{\infty} \lambda\, p(\lambda)\, d\lambda\, /\, \int_{0}^{\infty} p(\lambda)\, d\lambda$. We use these metrics in Fig. C.5 and Fig. C.6.

Note that measuring loss landscapes and Hessian eigenvalues without considering a regularization on clean datasets would lead to incorrect results, since NN training optimizes $\ell_2$ regularized NLL on augmented training datasets—not NLL on clean training datasets. We visualize loss landscapes and Hessian eigenvalues with respect to *"$\ell_2$ regularized NLL loss"* on *"augmented training datasets"*.

**Fourier analysis of feature maps.** We analyze feature maps in Fourier space to demonstrate that MSA is a low-pass filter as shown in Fig. 2, Fig. 8, and Fig. D.2. Fourier transform converts feature maps into frequency domain. We represent these converted feature maps on normalized frequency domain, so that the highest frequency components are at $f = \{-\pi, +\pi\}$, and the lowest frequency components are at $f = 0$. We mainly report the amplitude ratio of high-frequency components and low-frequency components by using $\Delta$ log amplitude, the difference in log amplitude at $f = \pi$ and $f = 0$. Yin et al. (2019) also analyzed the robustness of NNs from a Fourier perspective, but their research focused on input images—not feature maps—in Fourier spaces.

## B  MSAs Behave Like Spatial Smoothings

As mentioned in Section 1.1, spatial smoothings before subsampling layers help CNNs see better (Zhang, 2019; Park & Kim, 2021). Park & Kim (2021) showed that such improvement in performance is possible due to *spatial ensembles* of feature map points. To this end, they used *the (Bayesian) ensemble average of predictions for proximate data points* (Park et al., 2021), which exploits *data uncertainty* (i.e., a distribution of feature maps) as well as model uncertainty (i.e., a posterior probability distribution of NN weights):

$$p(\boldsymbol{z}_j | \boldsymbol{x}_j, \mathcal{D}) \simeq \sum_i \pi(\boldsymbol{x}_i | \boldsymbol{x}_j)\, p(\boldsymbol{z}_j | \boldsymbol{x}_i, \boldsymbol{w}_i) \tag{2}$$

where $\pi(\boldsymbol{x}_i | \boldsymbol{x}_j)$ is the normalized importance weight of a feature map point $\boldsymbol{x}_i$ with respect to another feature map point $\boldsymbol{x}_j$, i.e., $\sum_i \pi(\boldsymbol{x}_i | \boldsymbol{x}_j) = 1$. This importance is defined as the similarity between $\boldsymbol{x}_i$ and $\boldsymbol{x}_j$. $p(\boldsymbol{z}_j | \boldsymbol{x}_i, \boldsymbol{w}_i)$ and $p(\boldsymbol{z}_j | \boldsymbol{x}_j, \mathcal{D})$ stand for NN prediction and output predictive distribution, respectively. $\boldsymbol{w}_i$ is the NN weight sample from the posterior $p(\boldsymbol{w} | \mathcal{D})$ with respect to the training dataset $\mathcal{D}$. Put shortly, Eq. (2) spatially complements a prediction with other predictions based on similarities between data points. For instance, a $2 \times 2$ box blur spatially ensembles four neighboring feature map points, each with ¼ of the same importance.

We note that the formulations for self-attention and the ensemble averaging for proximate data points are identical. The Softmax term and $\boldsymbol{V}$ in Eq. (1) exactly correspond to $\pi(\boldsymbol{x}_i | \boldsymbol{x}_j)$ and $p(\boldsymbol{z}_j | \boldsymbol{x}_i, \boldsymbol{w}_i)$ in Eq. (2). The weight samples in Eq. (2) is correspond to the multi-heads of MSAs (See also (Hron et al., 2020)).

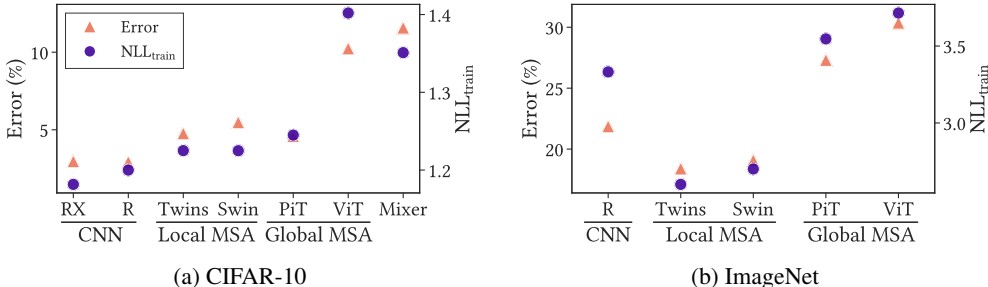

(a) CIFAR-10           (b) ImageNet

Figure C.1: **The lower the training NLL, the lower the test error.** "R" is ResNet and "RX" is ResNeXt. *Left:* In small data regimes, such as CIFAR-10 and CIFAR-100 (Fig. 5a), the cons of MSAs outweigh their pros; i.e., the non-convex losses disturb ViT optimization. *Right:* Large datasets convexify the loss functions. Therefore, the pros of MSAs outweigh their cons in large data regimes; i.e., MSAs help NNs learn strong representations by flattening the loss landscapes.

Likewise, the properties of spatial smoothing are the same as those of MSAs (Park & Kim, 2021): ① Spatial smoothing improves the accuracy of CNNs. In addition, spatial smoothing predicts well-calibrated uncertainty. ② Spatial smoothing is robust against MC dropout (which is equivalent to image occlusion), data corruption, and adversarial attacks, and particularly robust against high-frequency noise. ③ Spatial smoothing layers closer to the output layer significantly improves the predictive performance. In addition, concurrent works suggest that MSA blocks behave like a spatial smoothing. Wang et al. (2022) provided a proof that `Softmax`-normalized matrix is a low-pass filter, although this does not guarantee that MSA *blocks* will behave like low-pass filters. Yu et al. (2021b) demonstrated that the MSA layers of ViT can be replaced with average pooling layers.

Taking all these observations together, *we provide an explanation of how MSAs work by addressing themselves as a general form of spatial smoothing or an implementation of ensemble averaging for proximate data points.* Spatial smoothing improves performance in the following ways (Park & Kim, 2021): ❶ Spatial smoothing helps in NN optimization by flattening the loss landscapes. Even a small 2×2 box blur filter significantly improves performance. ❷ Spatial smoothing is a low-pass filter. CNNs are vulnerable to high-frequency noises, but spatial smoothing improves the robustness against such noises by significantly reducing these noises. ❸ Spatial smoothing is effective when applied at the end of a stage because it aggregates all transformed feature maps. This paper empirically shows that these mechanisms also apply to MSAs.

## C  ViTs From A Loss Landscape Perspective

This section provides further explanations of the analysis in Section 2.

**The lower the NLL on the training dataset, the lower the error on the test dataset.** Figure 5a demonstrates that low training NLLs result in low test errors on CIFAR-100. The same pattern can be observed on CIFAR-10 and ImageNet as shown in Fig. C.1.

In small data regimes, such as CIFAR-10 (Fig. C.1a) and CIFAR-100 (Fig. 5a), both the error and the NLL$_{\text{train}}$ of ViTs are inferior to those of CNNs. This suggests that the cons of MSAs outweigh their pros. As discussed in Fig. 4, ViTs suffers from the non-convex losses, and these non-convex losses disturb ViT optimization.

In large data regimes, such as ImageNet (Fig. C.1b), both the error and the NLL$_{\text{train}}$ of ViTs with local MSAs are superior to those of CNNs. Since large datasets convexify the loss functions as discussed in Fig. 4, the pros of MSAs outweigh their cons. Therefore, MSAs help NNs learn strong representations by flattening the loss landscapes.

**Rigorous discussion on the regularization of CNN's inductive bias.** In Fig. 5a, we compare models of similar sizes, such as ResNet-50 and ViT-Ti. Through such comparison, we show that a weak inductive bias hinders NN training, and that inductive biases of CNNs—inductive bias of Convs and multi-stage architecture—help NNs learn strong representations. However, inductive biases of

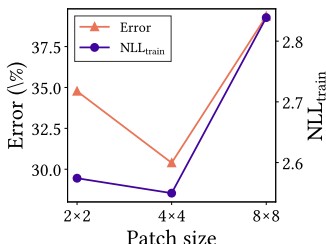 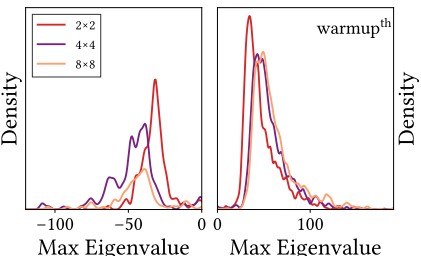

(a) Error and NLL$_{train}$ of ViT for patch size

(b) Negative and positive Hessian max eigenvalue spectra in early phase of training

Figure C.2: **A small patch size does not guarantee better performance**. We analyze ViTs with three embedded patch sizes: $2 \times 2$, $4 \times 4$, and $8 \times 8$. Note that every MSA has a global receptive fields. *Left:* As expected, a large patch size harms the performance, but surprisingly, the same is observed from a small patch size. *Right:* A small patch size, or a weak inductive bias, produces negative eigenvalues. This is another evidence that a weak inductive bias hinders NN optimization. On the other hand, MSAs with a small patch size reduce the magnitude of eigenvalues because they ensemble a large number of feature map points. Performance is optimized when these two effects are balanced.

CNNs produce better test accuracy for the same training NLL, i.e., Convs somewhat regularize NNs. We analyze two comparable models in terms of NLL$_{train}$ on CIFAR-100. The NLL$_{train}$ of ResNet-18, a model smaller than ResNet-50, is 2.31 with an error of 22.0%. The NLL$_{train}$ of ViT-S, a model larger than ViT-Ti, is 2.17 with an error of 30.4%. In summary, the inductive biases of CNNs improve accuracy for similar training NLLs.

Most of the improvements come from the multi-stage architecture, not the inductive bias of Convs. The NLL$_{train}$ of the PiT-Ti, a multi-stage ViT-Ti, is 2.29 with an error of 24.1 %. The accuracy of PiT is only 1.9 percent point lower than that of ResNet. In addition, the small receptive field also regularizes ViT. See Fig. 7.

**ViT does not overfit a small training dataset even with a large number of epochs.** Figure 5b shows that ViT does not overfit small training datasets, such as CIFAR. The same phenomenon can be observed in ViT training with a large number of epochs.

In Fig. C.3, we train ViT and ResNet for 75, 150, 300, 600, and 1200 epochs. Results show that both NLL$_{train}$ and error decrease as the number of epochs increases. The predictive performances of ViT are inferior to those of ResNet across all ranges of epochs.

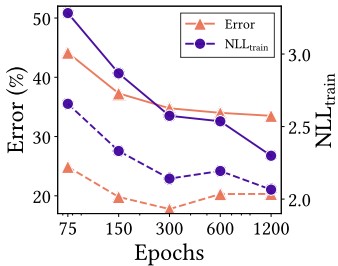

Figure C.3: **A large number of epochs does not make ViT overfit the training dataset of CIFAR**. Solid line is the predictive performance of ViT and dashed line is that of ResNet.

**A smaller patch size does not always imply better results.** ViT splits image into multiple patches. The smaller the patch size, the greater the flexibility of expression and the weaker the inductive bias. By analyzing ViT with three patch sizes—$2 \times 2$, $4 \times 4$, and $8 \times 8$—we demonstrate once again that a weak inductive bias disturbs NN optimization.

Figure C.2a shows the error on the test dataset and the NLL on the training dataset of CIFAR-100. As expected, a large patch size harms the performance on both datasets. Surprisingly, however, a small patch size also shows the same result. As such, appropriate patch sizes help ViT learn strong representations and do not regularize ViT.

The Hessian max eigenvalue spectra in Fig. C.2b explain this observation. Results reveal that a small patch size reduces the magnitude of Hessian eigenvalues but produces negative Hessian eigenvalues. In other words, the weak inductive bias makes loss landscapes flat yet non-convex. A large patch size suppresses negative eigenvalues. On the other hand, it not only limits the model expression but also sharpens loss landscapes. Performance is optimized when these two effects are balanced.

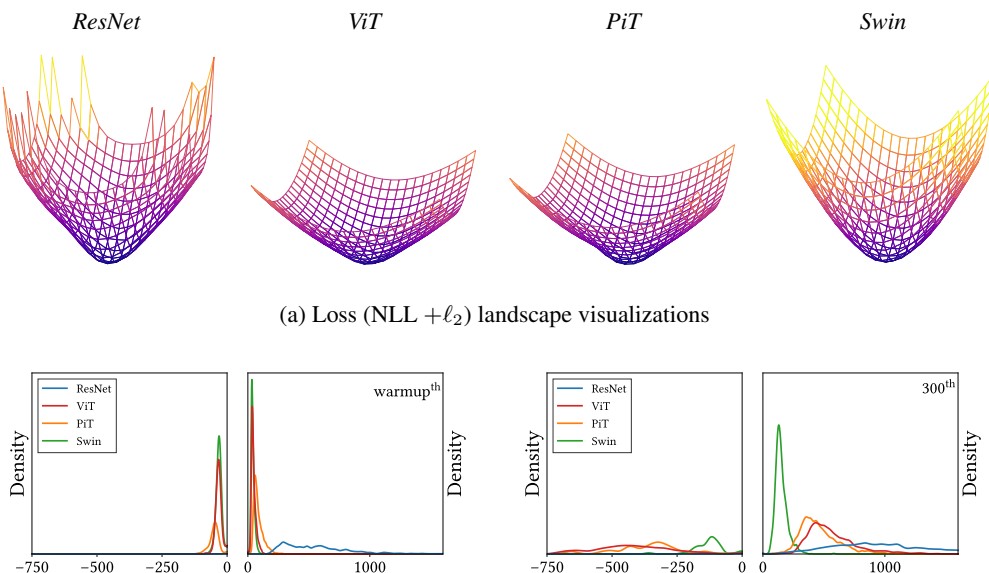

(a) Loss (NLL $+\ell_2$) landscape visualizations

(b) Negative and positive Hessian max eigenvalue spectra in early phase (*left*) and late phase (*right*) of training

Figure C.4: **A multi-stage architecture (in PiT) and a local MSA (in Swin) also flatten the loss landscapes.** *Top:* PiT has a flatter loss landscape than ViT near the optimum. Swin has an almost perfectly smooth parabolic loss landscape, which leads to better NN optimization. *Bottom:* A multi-stage architecture in PiT suppresses negative Hessian eigenvalues. A local MSA in Swin produces negative eigenvalues, but significantly reduces the magnitude of eigenvalues.

**A multi-stage architecture in PiT and a local MSA in Swin also flatten loss landscapes.** As explained in Fig. 1, an MSA smoothens loss landscapes. Similarly, a multi-stage architecture in PiT and local MSA in Swin also help NN learn strong representations by smoothing the loss landscapes.

Figure C.4 provides loss landscape visualizations and Hessian eigenvalue spectra of ResNet, ViT, PiT, and Swin. Figure C.4a visualizes the global geometry of the loss functions. The loss landscapes of PiT is flatter than that of ViT near the optimum. Since Swin has more parameters than ViT and PiT, $\ell_2$ regularization determines the loss landscapes. All the loss surfaces of ViTs are smoother than that of ResNet. Figure C.4b shows the local geometry of the loss functions by using Hessian eigenvalues. In the early phase of training, a multi-stage architecture in PiT helps training by suppressing negative Hessian eigenvalues. A local MSA in Swin produces negative eigenvalues, but significantly reduces the magnitude of eigenvalues. Moreover, the magnitude of Swin's Hessian eigenvalue does not significantly increases in the late phase of learning.

**A lack of heads may lead to non-convex losses.** Neural tangent kernel (NTK) (Jacot et al., 2018) theoretically implies that the loss landscape of a ViT is convex and flat when the number of heads or the number of embedding dimensions per head goes to infinity (Hron et al., 2020; Liu et al., 2020). In particular, Liu et al. (2020) suggests that $||H|| \simeq \mathcal{O}(1/\sqrt{m})$ where $||H||$ is the Hessian spectral norm and $m$ is the number of heads or the number of embedding dimensions per head. Therefore, in practical situations, insufficient heads may cause non-convex and sharp losses.

Fig. C.5 empirically show that a lot of heads in MSA convexify and flatten the loss landscapes (cf. Michel et al. (2019)). In this experiment, we use NEP and APE to measure the non-convexity and the sharpness as introduced in Appendix A.2. Results show that both NEP and APE decrease as the number of heads increases. Likewise, Fig. C.6 shows that high embedding dimensions per head also convexify and flatten losses. The exponents of APE are $-0.562$ for the number of heads and $-0.796$ for the number of embedding dimensions, which are in close agreement with the value predicted by the theory of $-1/2$.

**Large models have a flat loss in the early phase of training.** Figure C.7 analyzes the loss landscapes of large models, such as ResNet-101 and ViT-S. As shown in Fig. C.7a, large models explore

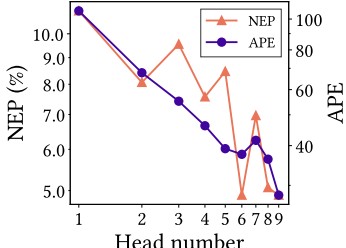
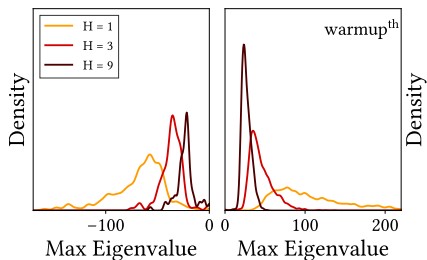

(a) NEP (non-convexity) and APE (sharpness) of head number

(b) Hessian negative and positive max eigenvalue spectra in early phase of training

Figure C.5: **Multi-heads convexify and flatten loss landscapes**. *Left:* We use negative max eigenvalue proportion (NEP) and average of positive max eigenvalues (APE) to quantify, respectively, the non-convexity and sharpness of loss landscapes. As the number of heads increases, loss landscapes become more convex and flatter. *Right:* Hessian max eigenvalue spectra also show that multi-head suppress negative eigenvalues and reduce the magnitude of eigenvalues.

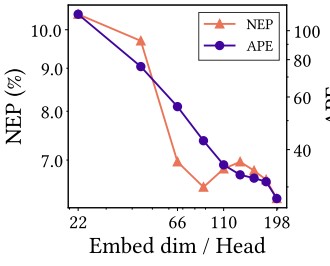
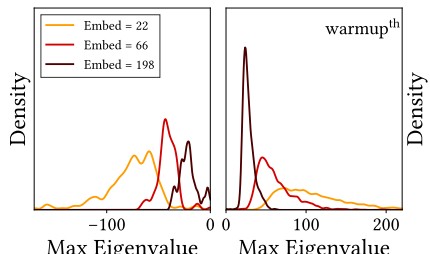

(a) NEP (non-convexity) and APE (sharpness) of embedding dim

(b) Hessian negative and positive max eigenvalue spectra in early phase of training

Figure C.6: **High embedding dimensions per head convexify and flatten the loss landscape**. *Left:* As the number of embedding dimensions per head increases, loss landscapes become more convex and flat. *Right:* Hessian max eigenvalue spectra also show that high embedding dimensions suppress negative eigenvalues and reduce the magnitude of eigenvalues as shown in Fig. C.5.

low NLLs. This can be a surprising because loss landscapes of large models are globally sharp as shown in Fig. C.7b.

The Hessian eigenvalue spectra in Fig. C.7c provide a solution to the problem: Hessian eigenvalues of large models are smaller than those of small models in the early phase of training. This indicates that large models have flat loss functions locally.

# D VITS FROM A FEATURE MAP PERSPECTIVE

This section provides further explanations of the analysis in Section 3 and Section 4.1.

**MSAs in PiT and Swin also ensemble feature maps.** In Fig. 9, we show that MSAs in ViT reduce feature map variances. The same pattern can be observed in PiT and Swin. Figure D.1 demonstrates that MSAs in PiT and Swin also reduce the feature map variances, suggesting that they also ensemble feature maps. One exception is the 3rd stage of Swin. MSAs suppresses the increase in variance at the beginning of the stage, but not at the end of the stage.

**MSAs in PiT and Swin are also low-pass filters.** As discussed in Fig. 8, MSAs in ViTs are low-pass filters, while MLPs in ViT and Convs in ResNet are high-pass filters. Likewise, we demonstrate that MSAs in PiT and Swin are also low-pass filters.

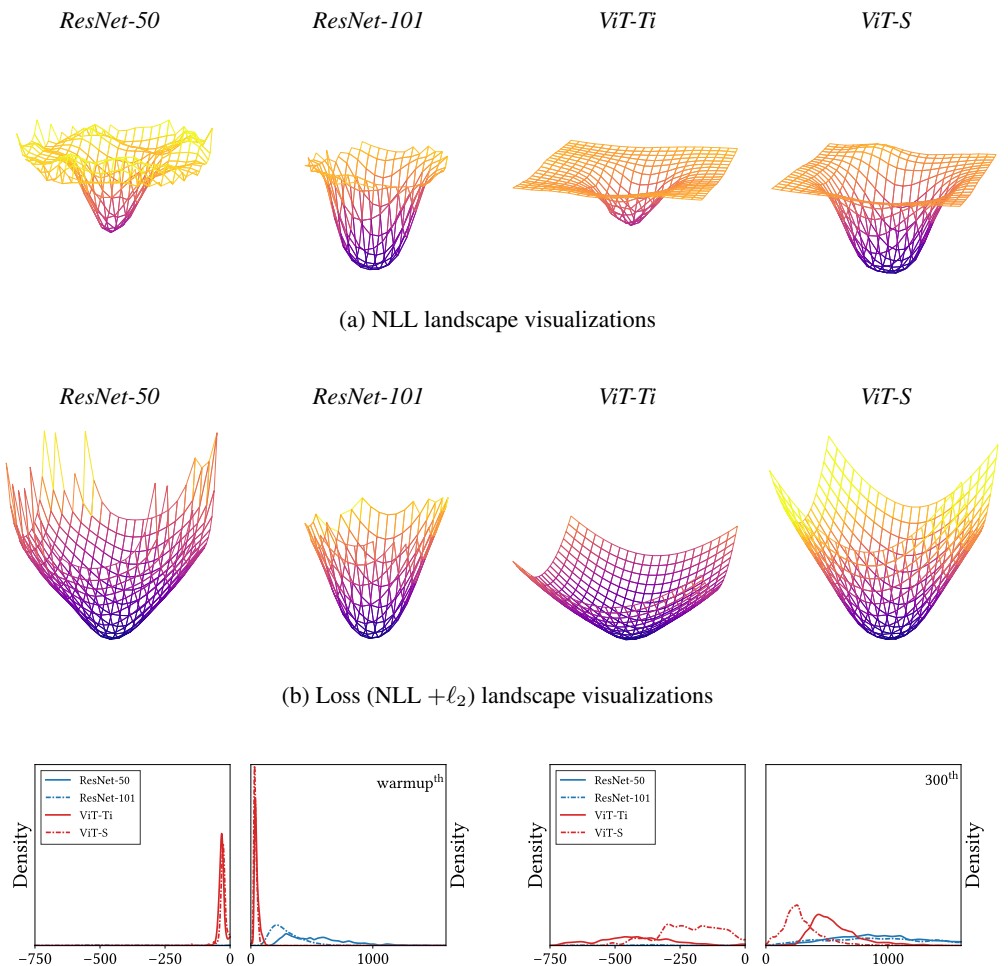

(a) NLL landscape visualizations

(b) Loss (NLL $+\ell_2$) landscape visualizations

(c) Negative and positive Hessian max eigenvalue spectra in early phase (*left*) and late phase (*right*) of training

Figure C.7: **Loss landscapes of large models.** ResNet-50 and ResNet-101 are comparable to ViT-Ti and ViT-S, respectively. *Top:* Large models explore low NLLs. *Middle:* Loss landscape visualizations show that the global geometry of large models is sharp. *Bottom:* The Hessian eigenvalues of large models are smaller than those of small models. This suggests that large models have a flat local geometry in the early phase of training, and that this flat loss helps NNs learn strong representations. In the late phase of training, large ViTs have flat minima while large ResNet has a sharp minimum.

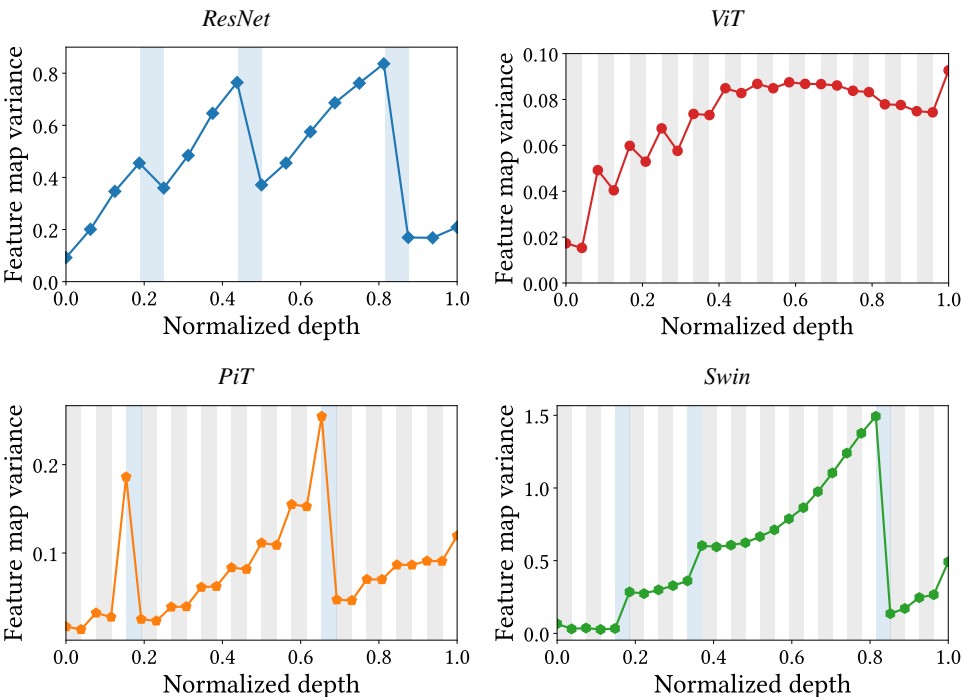

Figure D.1: **MSAs in PiT and Swin also reduce feature map variance** except in 3$^{rd}$ stage of Swin. White, gray, and blue areas are Conv/MLP, MSA, and subsampling layers, respectively.

Figure D.2 shows the relative log amplitude of Fourier transformed feature maps. As in the case of ViT, MSAs in PiT and Swin generally decrease the amplitude of high-frequency signals; in contrast, MLPs increases the amplitude.

**Multi-stage ViTs have a block structures.** Feature map similarities of CNNs shows a block structure (Nguyen et al., 2021). As Raghu et al. (2021) pointed out, ViTs have a uniform representations across all layers. By investigating multi-stage ViTs, we demonstrate that subsampling layers create a characteristic block structure of the representation. See Fig. D.3.

**Convs at the beginning of a stage and MSAs at the end of a stage play an important role.** Figure D.4 shows the results of a lesion study for ResNet and ViTs. In this experiment, we remove one $3 \times 3$ Conv layer from the bottleneck block of a ResNet, and one MSA or MLP block from ViTs. Consistent results can be observed for all models: Removing Convs at the beginning of a stage and MSAs at the end of a stage significantly harm accuracy. As a result, the accuracy varies periodically.

# E  EXTENDED INFORMATION OF ALTERNET

This section provides further informations on AlterNet.

**Detailed architecture of AlterNet.** Section 4 introduces AlterNet to harmonize Convs with MSAs. Since most MSAs take pre-activation arrangements, pre-activation ResNet is used as a baseline for consistency. We add one CNN block to the last stage of ResNet to make the number of blocks even. A local MSA with relative positional encoding from Swin is used for AlterNet. However, for simplicity of implementation, we do not implement detailed techniques, such as a cyclic shift and layer-specific initialization. For CIFAR, the patch size of the MSA is $1 \times 1$ and the window size is $4 \times 4$. If all Conv blocks are alternately replaced with MSA, AlterNet becomes a Swin-like model.

In order to achieve better performance, NNs should strongly aggregate feature maps at the end of models as discussed in Section 3 and Section 4. To this end, AlterNet use 3, 6, 12, 24 heads for MSAs in each stage.

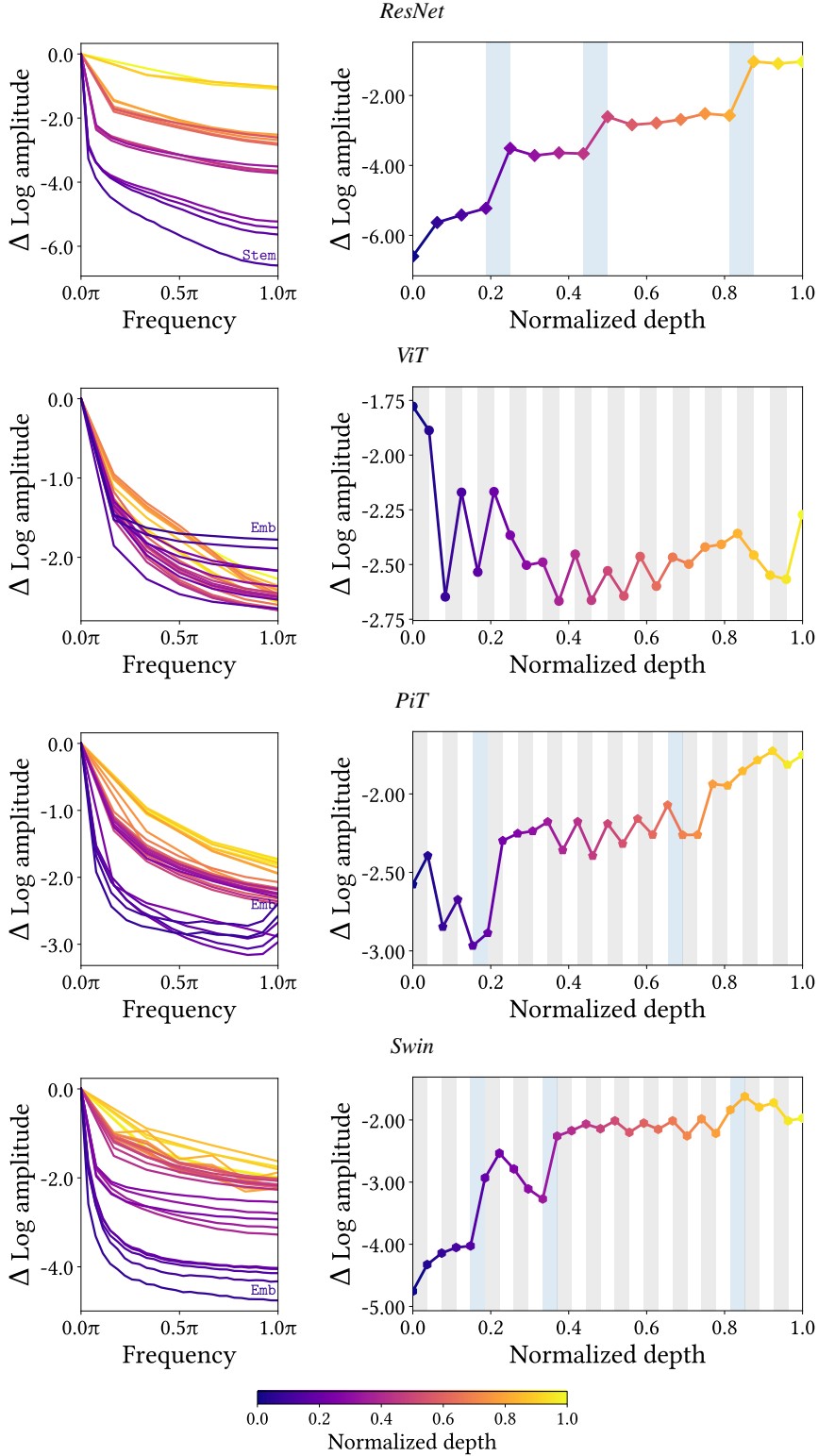

Figure D.2: **MSAs in PiT and Swin also reduce high-frequency signals**. *Left:* $\Delta$ log amplitude of Fourier transformed feature map. We only provide the diagonal components. *Right:* The high-frequency ($1.0\pi$) $\Delta$ log amplitude. White, gray, and blue areas are Conv/MLP, MSA, and subsampling layers, respectively.

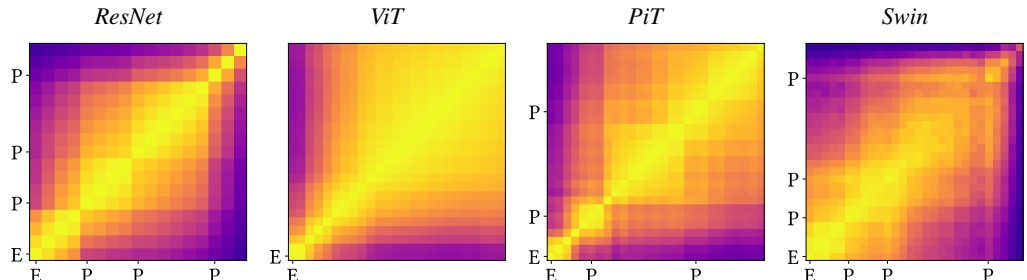

Figure D.3: **Multi-stage ViTs have block structures in representational similarities**. Block structures can be observed in all multi-stage NNs, namely, ResNet, PiT, and Swin. "E" is the stem/embedding and "P" is the pooling (subsampling) layer.

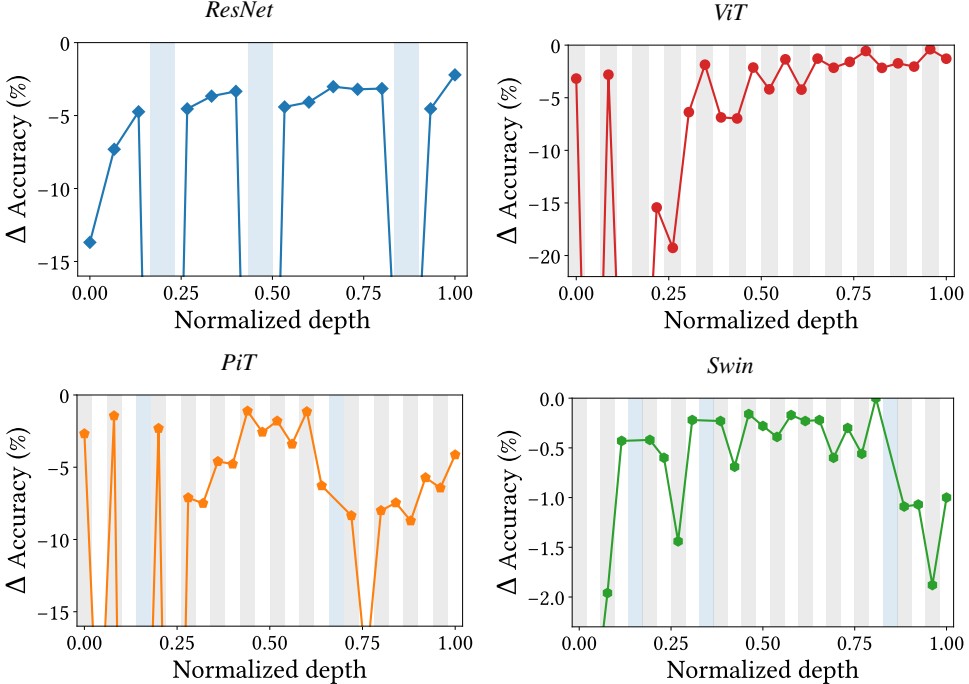

Figure D.4: **Lesion study shows that Convs at the beginning of a stage and MSAs at the end of a stage are important for prediction**. We measure the decrease in accuracy after removing one unit from the trained model. In this experiment, we can observe that accuracy changes periodically. The white, gray, and blue areas are Convs/MLPs, MSAs, and subsampling layers, respectively.

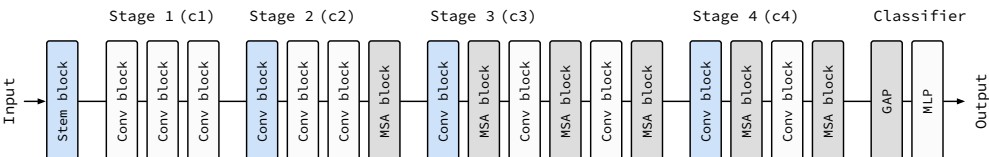

Figure D.5: **Detailed architecture of Alter-ResNet-50 for ImageNet-1K.** The white, gray, and blue blocks each represent Convs, MSAs, and subsampling blocks. This model alternately replaces Conv blocks with MSA blocks from the end of a stage. Following Swin, MSAs in stages 1 to 4 use 3, 6, 12, and 24 heads, respectively. We use 6 MSA blocks for ImageNet since large amounts of data alleviates the drawbacks of MSA. See Fig. 11 for comparison with the model for CIFAR-100, which uses 4 MSA blocks.

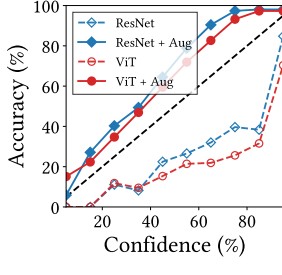 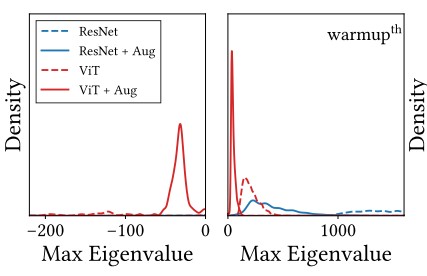

(a) Reliability diagram    (b) Hessian max eigenvalue spectrum

Figure F.1: **Distinctive properties of strong data augmentation.** "Aug" stands for strong data augmentation. *Left:* Strong data augmentation makes predictions underconfident on CIFAR-100. The same phenomenon can be observed on ImageNet-1K. *Right:* Strong data augmentation significantly reduces the magnitude of Hessian max eigenvalues. This means that the data augmentation helps NNs converge to better optima by flattening the loss landscapes. On the other hand, strong data augmentation produces a lot of negative Hessian eigenvalues, i.e., it makes the losses non-convex.

The computational costs of Conv blocks and MSA blocks are almost identical. The training throughput of Alter-ResNet-50 is 473 image/sec on CIFAR-100, which is 20% faster than that of pre-activation ResNet-50.

The optimal number of MSAs depends on the model and dataset, so we empirically determine the number of MSAs as shown in Fig. 12a. A large dataset allows a large number of MSAs. For ImageNet, we use 6 MSAs as shown in Fig. D.5, because a large datasets alleviates the shortcomings of MSAs.

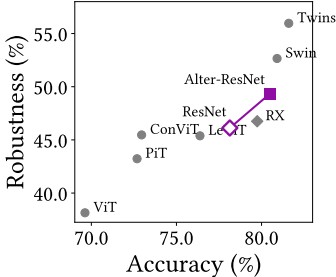

**MSAs improve the performance of CNNs on ImageNet.** Since MSAs complement Convs, MSAs improve the predictive performance of CNNs when appropriate build-up rules are applied as shown in Section 4.1. Figure E.1 illustrates the accuracy and robustness—mean accuracy on ImageNet-C—of CNNs and ViTs on ImageNet-1K. Since ImageNet is a large dataset, a number of ViTs outperform CNNs. MSAs with the appropriate build-up rules

Figure E.1: **MSA with the appropriate build-up rules significantly improves ResNet** on ImageNet. Robustness is mean accuracy on ImageNet-C. "RX" is ResNeXt.

significantly improves ResNet, and the predictive performance of AlterNet is on par with that of Swin in terms of accuracy without heavy modifications, e.g., the shifted windowing scheme (Liu et al., 2021). AlterNet is easy-to-implement and has a strong potential for future improvements. In addition, the build-up rules not only improve ResNet, but also other NNs, e.g., vanilla post-activation ResNet and ResNeXt; but we do not report this observation in order to keep the visualization simple.

## F  DISTINCTIVE PROPERTIES OF DATA AUGMENTATION

This section empirically demonstrates that NN training with data augmentation is different from training on large datasets. We compare DeiT-style strong data augmentation with weak data augmentation, i.e., resize and crop. In this section, "a result without data augmentation" stands for "a result only with weak data augmentation".

### F.1  DATA AUGMENTATION CAN HARM UNCERTAINTY CALIBRATION

Figure F.1a shows a reliability diagram of NNs with and without strong augmentation on CIFAR-100. Here, both ResNet and ViT without data augmentation (i.e., only with weak data augmentation) predict overconfident results. We show that strong data augmentation makes the predictive results underconfident (cf. Wen et al. (2021)). These are unexpected results because the predictions without data augmentation on large datasets, such as ImageNet, are not under-confident. A detailed investigation remains for future work.

## F.2  DATA AUGMENTATION REDUCES THE MAGNITUDE OF HESSIAN EIGENVALUES

How does data augmentation help an MSA avoid overfitting on a training dataset and achieve better accuracy on a test dataset? Figure F.1b shows the Hessian max eigenvalue spectrum of NNs with and without strong data augmentation. First of all, strong data augmentation reduces the magnitude of Hessian eigenvalues, i.e., data augmentation flattens the loss landscapes in the early phase of training. These flat losses leads to better generalization. On the other hand, strong data augmentation produces a lot of negative Hessian eigenvalues, i.e., data augmentation makes the losses non-convex. This prevents NNs from converging to low losses on training datasets. It is clearly different from the effects of large datasets discussed in Fig. 4—large datasets convexify the loss landscapes. A detailed investigation remains for future work.

