# OpenReview forum: "How Do Vision Transformers Work?"
_ICLR.cc/2022/Conference — ICLR 2022 Spotlight_

### Official Review · Reviewer_aDUi · 2021-11-02

**Correctness:** 4
**Technical Novelty And Significance:** 2
**Empirical Novelty And Significance:** 4
**Recommendation:** 8
**Confidence:** 3

**Main Review:**

Strenghts:

- Studies to better understand the behaviour of these models are valuable. The presented study is comprehensive in the sense that it includes many recent variants of ViTs and the explanations seem plausible.


Weaknesses:

- It is not clear to me how the construction rule for AlterNet is implemented in practice. The paper states "... does not improve performance sufficiently ...": on what dataset is this measured? Is this on a separate validation set or was this applied to the reported test set?

- It would have been great to see if the AlterNet modification can be universally applied, by showing thi on some additional base CNN models beyond ResNet-50.

- There are a couple of interesting findings and explanations, however, it seems hard to get to actionable design rules from them (beyond the proposed AlterNet). For example, the insight that loss smoothing methods aid in training seems to be generally true, not only for ViT, so it is not clear how much this insight really helps.

- It would be great to see a discussion on how the number of parameters influence the findings. Are the tested ViT variants and ResNets comparable in this respect? The main results are reported on "Tiny" variants of transformer architectures. Would some of the insights change with larger models?

Minor:

- Figure 10 (b): it seems that the ResNet figure is cut off from below.

**Summary Of The Paper:**

This paper attempts to explain the behaviour of multi-headed self-attention in vision transformers using a series of empirical observation. Some main observation are that the poor performance of ViT in the small data regime is not due to overfitting but to issues induced into the optimization procedure and that the behaviour of MSA is somewhat complementary to convolutions. Based on the observations a new model is proposed that combines convolution and MSAs. The experiments show that the new architecture improves results on smaller dataset.

**Summary Of The Review:**

I overall think that this paper shows some interesting phenomena, even though many of the observations are post-hoc and don't seem to lead to immediate actionable design rules. However, follow-up work may still benefit from these insights.

---

> ### Author Response · Authors · 2021-11-17
> **Author Responses to Reviewer aDUi**
>
> Thank you for your insightful feedback. We are pleased to hear that "follow-up work may still benefit from these insights". We address your concerns below.
>
> ---
>
> **Ⅳ-1. How the construction rule for AlterNet is implemented in practice.**
>
> In Fig. 12a, we apply the build-up rule to ResNet on the test set of CIFAR-100. We believe that the model optimized on CIFAR-100 (Fig. 11) works for other tasks as well. For example, the accuracy of the "Alter-ResNet for CIFAR-100" is 80.3% on ImageNet, which is a 2.3 percent point improvement over vanilla ResNet. The difference in accuracy between the model optimized on ImageNet (Fig. D.5) and this model is only 0.3 percent point.
>
>
> ---
>
> **Ⅳ-2. It would have been great to see if the AlterNet modification can be universally applied.**
>
> AlterNet modification can be applied to other CNN models. To demonstrate this, we also measured the performance of AlterNets based on post-activation ResNet and ResNeXt. Their accuracy improvements on CIFAR-100 are: 82.7% → 83.6% (post-activation ResNet) and 82.3% → 83.3% (ResNeXt). We did not report the results for the sake of visualization simplicity.
>
>
> ---
>
> **Ⅳ-3. It seems hard to get to actionable design rules from them (beyond the proposed AlterNet). For example, the insight that loss smoothing methods aid in training seems to be generally true, not only for ViT, so it is not clear how much this insight really helps.**
>
> Our findings have implications for future work, for example: (1) Since ViTs suffer from non-convex losses$^\dagger$, an optimizer that escapes the saddle points or an architecture modification to smoothen the local loss geometry may help improve the performance. In contrast, CNNs suffer from sharp losses$^\ddagger$. Therefore, techniques to flatten the loss landscape can help improve the performance; (2) Global average pooling (GAP) is a strong feature map aggregator for classification tasks. Therefore, we believe that MSA to be able to significantly improve the results in dense prediction tasks by ensembling feature maps, since NNs for dense prediction do not use GAP; (3) Since we have shown that MSA can improve performance in image classification on small datasets, we believe that this work opens up the possibility of applying MSA to other tasks in small data regimes. If our insights holds in other domains (that do not require long-range dependencies), this work can motivate the use of MSA in a variety of domains; and so on.
>
> ·
>
> ***UPDATED***) $\dagger$: Negative Hessian eigenvalues
>
> ***UPDATED***) $\ddagger$: Large positive Hessian eigenvalues
>
> ---
>
> **Ⅳ-4. Would some of the insights change with larger models?**
>
> All insights mentioned in the main text hold for larger models. One interesting observation not mentioned in the main text is that large ViT has a more convex and flatter loss landscape compared to small ViT *in the later phase of training*; in contrast, the loss landscape of large ResNet is sharper than that of small ResNet. Please refer to Fig. C.7.
>
> We analyze these phenomena via the loss landscape perspective. Additional experiments show that 'multi-heads' and 'high embedding dimensions per head (corresponding to the width)' of MSAs convexify and flatten the loss landscapes. Please refer to Fig C.5 and Fig C.6. The number of heads of a large ViT (e.g. ViT-S) is greater than that of a small ViT (e.g. ViT-Ti), but their depths are the same. In contrast, the depth of a large ResNet (e.g. ResNet-101) is higher than that of a small ResNet (e.g. ResNet-50), while their widths are the same.
>
> Furthermore, we tried to connect theory to the experimental results. From NTK [1, 2], we easily derive the following proposition:
>
> $$\vert \mathcal{O}(m^{-1/2}) - c_{\infty} \vert \leq \vert\vert H \vert\vert \leq  \mathcal{O}(m^{-1/2}) + c_{\infty}$$
>
> where $\vert\vert H \vert\vert$ is the Hessian spectral norm, $m$ is the number of heads, and $c_{\infty}$ is a small constant. Surprisingly, our empirical results shows that $\vert\vert H \vert\vert$  follows precise power-law scaling relation with the number of heads; and the exponent obtained from the experiments ( -0.562) is in close agreement with the value predicted by the proposition (-1/2). Please refer to Eq. 5 for more information. We leave a detailed investigation for future work.
>
> ·
>
> [1] Hron, Jiri, et al. "Infinite attention: NNGP and NTK for deep attention networks." ICML (2020).
>
> [2] Liu, Chaoyue, et al. "On the linearity of large non-linear models: when and why the tangent kernel is constant." NeurIPS (2020).

---

### Official Review · Reviewer_GbX6 · 2021-11-03

**Correctness:** 2
**Technical Novelty And Significance:** 3
**Empirical Novelty And Significance:** 2
**Recommendation:** 5
**Confidence:** 4

**Main Review:**

The paper shows that MSA is low-pass while convs are high-pass, thus they complement each other. The paper interprets deep models as a series of independent blocks, and then proposes a new architecture, in which the last conversation block is replaced with multi-head self-attention. The architecture called AlterNet, shows better performance and robustness on ImageNet and CIFAR-100 datasets.

Paper says “local MSAs with a 3 × 3 receptive field outperforms global MSA”. Can the authors please provide a relevant reference to a ViT work, which does not encapsulate patch-level interactions (with MSAs) altogether, and only relies on local within-patch attention? I would be amused to see a deep architecture with a receptive field as small as 3*3.

While the paper makes a good effort in explaining how ViTs work. Some of the findings and explanations might not be in line with existing literature e.g., Fig.1 of the paper https://arxiv.org/pdf/2106.01548.pdf shows that ViT landscape is not flattened, and is sharper.

The authors mention that "Multi-stage architecture in PiT and local MSA in Swin also flatten the loss landscapes” I think it is the hierarchial features, captured in PiT and Swin, that help them compared with vanilla ViTs where feature hierarchies are missing. CNNs also have feature hierarchies with subsequent pooling. I guess vanilla ViTs, PiT, Swin or CNNs, are all multi-stage. It is multi-scale feature hierarchies that is absent in vanilla ViTs, and is present in PiT, Swin, ResNets.  Also, Vanilla ViTs (based upon the standard attention is all you need architecture) do not capture local relationships within small image patches. Recent variants of ViTs which capture such local cues with dilated convolutions, or local self-attention, basically, introduce better inductive biases, since local cues are important for images.

**Summary Of The Paper:**

The authors show that the robustness and better performance of ViTs is attributed to their property of flattening the loss surface. Insights are taken from a parallel ICLR submission, where spatial smoothing has been shown to smooth loss landscape, and help improve robustness. The paper shows that the properties of spatial smoothing are exhibited by MSA as well.



**Summary Of The Review:**

- The visualized loss landscape in the paper is different from existing literature https://arxiv.org/pdf/2106.01548.pdf; where a sharpness aware optimizer is developed to flatten the loss landscape of ViTs.

- I am not entirely convinced that the findings in the paper fully explain how ViTs work. I guess, the paper draws analogy between MSA and spatial smoothing, and then builds most of its argument around that. An appropriate title reflecting that will be better, in my opinion.

- Some of the terminologies used in the paper confused me, e.g., multi-stage vs multi-scale. All deep architectures are multi-stage in my understanding. Most have multi-scale feature hierarchies. Vanilla ViTs have single-scale features throughout all stages.

- Small patches improve performance (counter to what is shown in the paper). See https://arxiv.org/pdf/2106.09681.pdf

---

> ### Author Response · Authors · 2021-11-17
> **Author Responses to Reviewer GbX6**
>
> Thank you for your insightful feedback. We address all of your concerns below. If you find our responses adequate, we would appreciate it if you consider increasing your score.
>
> ---
>
> **Ⅲ-1. The visualized loss landscape in the paper is different from Fig 1 of [1].**
>
> The difference between the loss landscape visualization in [1] and our results is due to the following three aspects: (1) They only visualize cross-entropy (NLL) landscape, while we visualize loss (NLL + $\ell_{2}$ regularization) landscape. Since NN training optimizes NLL + $\ell_{2}$—not NLL—we believe that it is appropriate to visualize NLL + $\ell_{2}$$^\dagger$; (2) They use training configurations that is significantly different from standard practice, while we use a DeiT-style configuration [2]. Since DeiT-style configuration is the de facto standard in ViT training (See, e.g. PiT, Swin, CoAtNet ⋯), we believe our insights can be applied to a larger number of studies; (3) They compare ViT-B (#Param: 87M) and ResNet-152 (#Param: 60M). In general, ViT is parameter-efficient but computation-inefficient—e.g., the parameter size of ViT-S is smaller than that of ResNet-101, but it is three times slower in terms of throughput on CIFAR. ViT shows a similar trend on ImageNet [1]. Therefore, we believe that it is better to compare ViT with 'ResNet with the same or smaller parameter size'.
>
> We provide not only loss landscape visualization, but also learning trajectory and Hessian spectra, a set of Hessian eigenvalues. These three different aspects consistently show that MSAs flatten loss landscapes$^\ddagger$. In addition, it explains how ViT is more robust against data corruptions and adversarial attacks than CNNs [3]. For a more information, please refer to Figure C.7.
>
> ·
>
> [1] Chen, Xiangning, et al. "When Vision Transformers Outperform ResNets without Pretraining or Strong Data Augmentations." arXiv:2106.01548 (2021).
>
> [2] Touvron, Hugo, et al. "Training data-efficient image transformers & distillation through attention." ICML (2021).
>
> [3] Naseer, Muzammal, et al. "Intriguing Properties of Vision Transformers" NeurIPS (2021).
>
> ·
>
> ***UPDATED***) $\dagger$: See also Fig 1 in https://arxiv.org/abs/1802.10026 and Fig 3 in https://arxiv.org/abs/1803.05407
>
> ***UPDATED***) $\ddagger$: Hessian of AlterNet (Fig 12c) also shows that MSAs flatten loss landscapes.
>
> ---
>
> **Ⅲ-2. In [4], small patches improve performance (counter to what is shown in the paper).**
>
> As discussed in Fig. 4, in large data regimes, the negative Hessian eigenvalues—the disadvantage of MSA due to the weak inductive bias—disappears, and only the advantage remains. [4] conducted experiments on ImageNet, so a fairly small patch does not harm the performance in that case (It may not be feasible to train a ViT with a patch size smaller than 8 × 8 on ImageNet).
>
> ·
>
> [4] El-Nouby, Alaaeldin, et al. "XCiT: Cross-Covariance Image Transformers." arXiv:2106.09681 (2021).
>
> ---
>
> **Ⅲ-3. Some of the terminologies used in the paper confused me, e.g., multi-stage vs multi-scale.**
>
> In our paper, "multi-stage NNs" stands for NNs including subsampling layers (See, e.g., [5]). To the best of our knowledge, the term "multi-scale" is mainly used to refer to different sizes of input data [6], or when NN aggregates feature maps of different scales [7]. Please correct us if we have misunderstood the terminologies.
>
> We did not deeply understand the last paragraph of "Main Review" (which starts with "The authors mention that ⋯"). We guessed that the paragraph was concerned about confusing terminologies. Please correct us if we have misunderstood the concern.
>
> ·
>
> [5] Mao, Xiaofeng, et al. "Towards Robust Vision Transformer." arXiv:2105.07926 (2021).
>
> [6] Redmon, Joseph, and Ali Farhadi. "YOLO9000: better, faster, stronger." CVPR (2017).
>
> [7] Lin, Tsung-Yi, et al. "Feature pyramid networks for object detection." CVPR (2017).
>
> ---
>
> **Ⅲ-4. I would be amused to see a deep architecture with a receptive field as small as 3*3.**
>
> First of all, we would like to clarify that the performance of the ViT is optimized when the size of receptive field is neither too large nor too small in Fig. 7, although local MSAs with a 3 × 3 receptive field outperforms global MSA; the optimal kernel size is 5 × 5 in this experiment on CIFAR. A large size of receptive field has advantages and disadvantages. A large receptive field stabilizes the loss landscape by ensembling a large number of feature maps. On the other hand, its over-fexibility makes the loss landscape non-convex. The performance is optimized when the two effects are balanced.
>
> As discussed in Ⅲ-2, large datasets mitigate the disadvantages of large receptive fields. As a result, the optimal size of the receptive field in large data regimes can be larger than that in small data regimes. In real-world situations, Swin's [8] receptive field is 7 × 7 on ImageNet.
>
> ·
>
> [8] Liu, Ze, et al. "Swin Transformer: Hierarchical Vision Transformer using Shifted Windows", ICCV (2021).

---

### Official Review · Reviewer_DmHc · 2021-11-03

**Correctness:** 4
**Technical Novelty And Significance:** 3
**Empirical Novelty And Significance:** 4
**Recommendation:** 8
**Confidence:** 4

**Main Review:**

+ The paper is very well reasoned and supports all arguments with solid experimental evidence. The presented story and explanations are both interesting, refreshingly different, and solidly argued.
+ The presented architecture is quite interesting, as it mixes many conv and MSA blocks, rather than a simple transition (i.e. in ViT-Hybrid or LeVIT, ...). Though the "Building-up rule" seems a bit arbitrary, and a more thorough reasoning would help the paper.
- The organization of the paper could be improved a bit. There is currently a bit of repetition within section 1 when talking about points (1)-(3) multiple times.
- Section 3 is a bit hand-wavy and relies on prior work for much of the argumentation. A similar line of though to section 2 would make the paper stronger.

Minor: Stylistically the numbered circles don't work all that well. Simple text 1., or (1) might work much better.

**Summary Of The Paper:**

This paper analyzes how vision transformers (and in contrast convolutional architectures) work from an optimization standpoints. The paper shows experimentally that contrary to popular belief ViTs do not directly benefit from their more expressive underlying representation, but instead smooth the loss landscape which aids training. The paper then presents a hybrid CNN-ViT architecture that combines some of the benefits of ViTs with advantages of CNNs.

**Summary Of The Review:**

I like this paper, and the insights it provides in terms of optimization landscapes of ViT vs CNN architectures. It also provides a pretty clear route towards better ViT or CNN architectures by combining the strengths of both. I wish though the experimental evaluation would focus more in ImageNet (currently mainly in appendix), and not CIFAR.

---

> ### Author Response · Authors · 2021-11-17
> **Author Responses to Reviewer DmHc**
>
> Thank you for your positive and insightful feedback. We are glad that you like our work and insights. Below, we address your concerns.
>
> ---
>
> **Ⅱ-1. Section 3 is a bit hand-wavy and relies on prior work for much of the argumentation.**
>
> We improved Section 3 by adding a proof that *self-attention is a low-pass filter*. Please refer to Proposition B.1 in Appendix B. The revision relies less on prior works. We will continue to revise and polish the manuscript during the discussion period.
>
>
> ---
>
> **Ⅱ-2. I wish though the experimental evaluation would focus more in ImageNet.**
>
> We are still working on ablation studies for AlterNet (a counterpart of Fig. 12a) since ViT training on ImageNet is computation-intensive. We will report the results.

---

### Official Review · Reviewer_Kvf7 · 2021-11-06

**Correctness:** 3
**Technical Novelty And Significance:** 4
**Empirical Novelty And Significance:** 4
**Recommendation:** 8
**Confidence:** 3

**Main Review:**

Pros:
- Paper focuses on addressing important problem and draws interesting insights
- Paper provides extensive empirical results and evidence
- Paper proposed a new Alter-ResNet architecture that is both more accurate and robust

Cons:
- Paper is generally hard to read and understand. Imperial results are reported in a variety of different forms and entirety of observations is hard to grasp. I read the paper carefully twice and I am still not 100% sure I understand ALL the intricacies of every experiment and how those unequivocally lead to insights drawn. The paper is just very dense. I believe the observations and insights are correct, the issue is manly expositions which could benefit from substantial improvements.
- The form of analysis conducted isn't well motivated or explained. This is perhaps the most negative aspect of the paper. Many different aspects of models are analyzed, e.g., NLL, Hassian max eigenvalue spectra, Frequency / Log Amplitude, etc. It is very hard to keep track of these different measures and little intuition is given for why these are ultimately "the right" measures to look at for a given empirical evaluation. Most experiments take the form (and I am simplifying of course) of "We did X, which means Y." However, no explanation is given for why X is an appropriate measure for Y; or explanation of potential alternative measures for Y and why those were not assess. Finally, no discussion of what optimal value or behavior of X would be appropriate for a good model and so on. I would suggest re-structured each of the empirical experiments to have clearer flow and argumentation.


**Summary Of The Paper:**

Paper presents empirical analysis Multi-headed Self-Attention (MSA) as part of Vision Transformer (ViT) and its variants. It identifies the following properties of MSA: (1) ViT with MSA improves both accuracy and generalization by flattening the loss landscape; (2) ViT with MSA behaves differently from Convolutional Neural Nets (CNNs). Specifically, ViT with MSA behaves as a low-pass and CNNs as high-pass filters. And (3) multi-stage neural nets behave like small independent models connected in series. Based on these observations, paper proposes a new architecture -- AlterNet, where CNN block is combined with MSA block and this "base" structure is replicated to produce deeper models. AlterNet is shown to outperform pure CNNs in both large and small data regimes.

**Summary Of The Review:**

Overall, the paper is empirically addressing a series of important questions about the (relatively) new class of architectures (ViT). The paper serves as both: (1) effectively a survey of recent papers and observations made in prior works and (2) empirical evaluation of these models. Empirical evaluations point to some interesting insights that are ultimately leveraged to build a new form of the model which is a (relatively simple) combination of the MSA and CNN.

The main concern with the paper is density of the text and the general lack of motivation for the various experimental designs, as well as sometimes somewhat tenuous (and not well explained) connections between empirical results and high-level insights. Despite these issues with exposition, which I think would benefit from fixing, I think the paper does address an important topic and drawn insights can be broadly useful. For these reasons I am in favor of acceptance.

---

> ### Author Response · Authors · 2021-11-17
> **Author Responses to Reviewer Kvf7**
>
>
>
> Thank you for your constructive and encouraging feedback. We are pleased to hear that "the paper does address an important topic and drawn insights can be broadly useful".
>
> Following the point that the "paper is generally hard to read and understand", we will continue to revise and polish the manuscript during the discussion period. For example, we improved readability by mentioning that training NLL is cross-entropy loss.

---

### Public Comment · ~Namuk_Park1 · 2022-04-05
**Revision**

We would like to thank all reviewers for their constructive suggestions. We revised the paper accordingly:

- We provided the preliminaries and detailed background information for the experiments in Appendix A.2. For example, we mentioned that loss landscapes and Hessian eigenvalues are visualized with respect to "*L2 regularized NLL loss*" on "*augmented training datasets*" because NNs are optimized with L2 regularization on augmented datasets.
- We provided additional experiments to show that 'multi-heads' and 'high embedding dimensions per head' of MSAs convexify and flatten the loss landscapes. Please refer to Fig. C.5 and Fig. C.6. Moreover, we tried to connect NTK theory to the experimental results.
- We revised and polished the manuscript to improve readability.

---

### Public Comment · ~Eugenie_Craig1 · 2022-08-20
**:)**

I want to know how do vision transformers works and I am glad I found my answer over here. When I was searching for your post online, I also found https://www.topessaywriting.org/samples/economics website on which I found a lot of essay samples. And it makes me happy when I found out that I can read them for free.

---

### Decision · Program_Chairs · 2022-01-20

**Decision:**

Accept (Spotlight)

**Comment:**

The paper presents an empirical analysis of Vision Transformers - and in particular multi-headed self-attention - and ConvNets, with a focus on optimization-related properties (loss landscape, Hessian eigenvalues). The paper shows that both classes of models have their strengths and weaknesses and proposes a hybrid model that takes the best of both worlds and demonstrates good empirical performance.

Reviewers are mostly very positive about the paper. Main pro is that analysis is important and this paper does a thorough job at it and draws some useful insights. There are several smaller issues with the presentation and the details of the content, but the authors did a good job addressing these in their responses.

Overall, it's a good paper on an important topic and I recommend acceptance.